# Model-Guided Microstimulation Steers Primate Visual Behavior

**Johannes Mehrer**[1,†], **Ben Lonnqvist**[1,4], **Anna Mitola**[2,3], **Paolo Papale**[2,*], **Martin Schrimpf**[1,*,†]
[1]EPFL, [2]Netherlands Institute for Neuroscience, [3]University of Parma, [4] Dandelion Science

## Abstract

Brain stimulation is a powerful tool for understanding cortical function and holds promise for therapeutic interventions in neuropsychiatric disorders. Initial visual prosthetics apply electric microstimulation to early visual cortex which can evoke percepts of simple symbols such as letters. However, these approaches are fundamentally limited by hardware constraints and the low-level representational properties of this cortical region. In contrast, higher-level visual areas encode more complex object representations and therefore constitute a promising target for stimulation — but determining representational targets that reliably evoke object-level percepts constitutes a major challenge. We here introduce a computational framework to causally model and guide stimulation of high-level cortex, comprising three key components: (1) a perturbation module that translates microstimulation parameters into spatial changes to neural activity; (2) topographic models that capture the spatial organization of cortical neurons and thus enable prototyping of stimulation experiments; and (3) a mapping procedure that links model-optimized stimulation sites back to primate cortex. Applying this framework in two macaque monkeys performing a visual recognition task, model-predicted stimulation experiments produced significant in-vivo changes in perceptual choices. Per-site model predictions and monkey behavior were strongly correlated, underscoring the promise of model-guided stimulation. Image generation further revealed a qualitative similarity between in-silico stimulation of face-selective sites and a patient's report of facephenes. This proof-of-principle establishes a foundation for model-guided microstimulation and points toward next-generation visual prosthetics capable of inducing more complex visual experiences.

## 1 Introduction

Vision is fundamental to human experience, enabling navigation, object recognition, and social interaction. For individuals with visual impairments, restoring even basic visual function could dramatically improve quality of life. Visual prosthetic devices represent a promising approach to bypass damaged tissue along the visual processing hierarchy (e.g. retina, optic nerve, lateral geniculate nucleus) and directly stimulate the visual cortex to shape or evoke visual percepts.

Current visual prosthetic approaches are in a prototypical development stage, but have already achieved remarkable successes: microstimulation of primate early visual areas can reliably evoke percepts of simple geometric shapes and even letters (Chen et al., 2020b; Beauchamp et al., 2020; Fernandez et al., 2021). These approaches to visual prosthetics rely on the spatial arrangement of neurons in early visual cortex, which mirrors the layout of the visual field: nearby points in the visual field and thus on the retina correspond to nearby points on cortex, a principle referred to as retinotopy (Hubel & Wiesel, 1962; 1968; Engel et al., 1997).

However, these approaches are fundamentally limited by the number of electrodes that can be implanted in early visual cortex, and by the representational properties of early visual areas: neurons in primary and secondary visual cortex (V1, V2) encode simple local features such as location or orientation of bars and simple combinations thereof (Hubel & Wiesel, 1962; 1965; 1968; Hegdé

---

*Equal supervision by P.P. and M.S.
†Correspondence: `[johannes.mehrer, martin.schrimpf]@epfl.chfs`

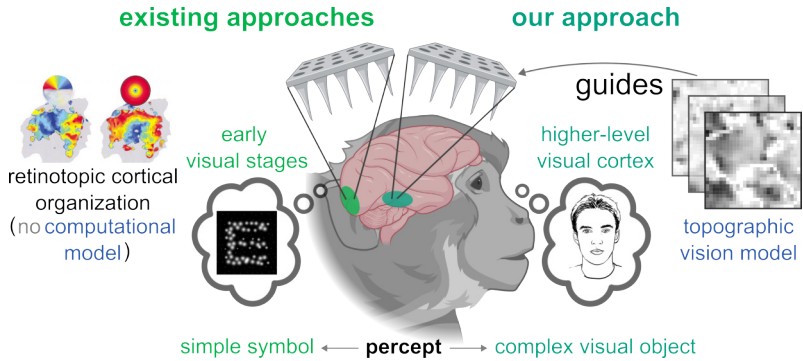

**Figure 1: Overview of approach.** Existing approaches to visual prosthetics microstimulate early cortical areas or even earlier parts of the visual processing hierarchy, do not use computational models for the selection of stimulation sites, and instead rely on retinotopic organization where nearby locations in the visual field are represented in nearby locations in the early visual system. These approaches have successfully been shown to elicit percepts of simple visual symbols such as letters, but are limited by the low-level representational properties of early visual regions. We propose a model-guided approach that targets higher-level visual cortex via computational simulations with the goal of eliciting percepts of complex visual objects.

& Van Essen, 2000; Anzai et al., 2007). Stimulation of these regions elicits elementary percepts such as phosphenes or simple shapes, and is currently not capable of evoking complex object-level representations required to restore rich visual experience. For individuals with profound visual impairments, the ability to perceive and recognize objects such as faces, tools, or scenes would open the possibility of richer and possibly more useful visual perception. Achieving this level of perceptual complexity might therefore require targeting cortical regions that explicitly encode complex visual objects — but effective stimulation of such high-level and complex representations remains an unsolved challenge.

Here, we propose to use computational models to guide microstimulation directly targeting higher-level visual cortex. Higher visual regions are known to underlie the representations of complex visual objects such as faces and scenes. However, the influence of retinotopy on object representation decreases strongly from early to higher-level visual regions (Issa & DiCarlo, 2012; Silson et al., 2015; Yue et al., 2020; Poltoratski et al., 2021). Thus, in higher-level visual regions, retinotopy is much less useful as a guiding principle for causal intervention techniques. Rather, the organization of higher-level regions is shaped by more complex visual and semantic features such as animacy vs. inanimacy and high-level category selectivity (Kriegeskorte et al., 2008; Kanwisher, 2017). These more abstract principles alone do not give clear guidance with respect to eliciting more complex visual percepts.

To address this challenge, we develop a model-guided approach to microstimulation in higher-visual cortex (**Fig. 1**). We present early successes of applying model predictions experimentally to two macaque monkeys. Specifically, we show that optimizing the combination of visual stimuli and stimulation parameters via simulations in brain-mapped topographic networks allows for predicting monkey visual behavioral responses in a complex object recognition task. Model-predicted changes to behavior are strongly correlated with actual experimentally observed changes in monkey behavior, although the model tends to overestimate the behavioral effect. Model-predicted experiments also lead to a substantial shift in monkey behavior along a target direction. To qualitatively interpret the effect of stimulation in-silico, we employ image generation on the simulated neural activity patterns during microstimulation and observe the emergence and enlargement of faces and face-features when stimulating in face-selective regions.

## 2 BACKGROUND & RELATED WORK

**Visual Cortex Stimulation.** A common implant for intracortical recording and stimulation in primates is the *Utah array*: a 96- or 64-channel microelectrode grid that allows simultaneous multi-site

recordings and the application of electrical pulse trains to focal patches of cortex. In early visual cortex, site selection and interpretation of stimulation are guided by retinotopy - a roughly point-to-point mapping from visual field locations to cortical locations, so that stimulating an electrode tends to evoke a phosphene at the receptive-field position represented beneath that electrode (Hubel & Wiesel, 1962; 1968).

Using retinotopy as orientation principle, existing prototypes of visual prostheses target sets of electrodes whose receptive fields tile desired visual-field positions and then induce static or dynamic stimulation patterns to elicit percepts of simple shapes or letters (Chen et al., 2020b; Beauchamp et al., 2020; Fernandez et al., 2021). Behavioral outcomes are typically quantified with forced-choice tasks that probe detection, localization, or identification, yielding psychometric functions over current amplitude, pulse rate, or stimulus strength and associated summary statistics such as thresholds or changes in area under the curve. While effective for low-level percepts, this retinotopy-based strategy is inherently constrained by electrode count and by the representational granularity of early areas, which primarily encode local features. As such, it does not naturally extend to evoking object-level percepts which are more closely associated with higher-level visual cortex.

**Models of the Brain.** Over the past decade, artificial neural networks (ANNs) have emerged as powerful system-level models of the visual brain that explain substantial variance in neural and behavioral responses (Yamins et al., 2014; Khaligh-Razavi & Kriegeskorte, 2014; Schrimpf et al., 2018; Mehrer et al., 2021; Gokce & Schrimpf, 2025). Recently, these models have been endowed with explicit cortical topography: *topographic* ANNs place units on a 2D sheet and are trained with spatial regularizers, yielding smoothly organized maps across layers (Lee et al., 2020; Keller et al., 2021; Lu et al., 2023; Margalit et al., 2024; Deb et al., 2025; Rathi & Mehrer et al., 2025). In deeper layers, they exhibit category-selective patches (Margalit et al., 2024) reminiscent of the functional organization of high-level visual cortex (Kanwisher, 2017; Tsao et al., 2003; 2006; Freiwald et al., 2009).

Because their representations are spatially embedded, topographic models can simulate the focal neural effects of currents applied via causal intervention techniques: localized perturbations can be applied to model tissue and the model can predict how the induced neural activation changes propagate across the simulated cortical sheet to predict downstream behavioral consequences. Prior work has evaluated such perturbation modules *offline*, showing that topographic models can anticipate the behavioral effects of different causal intervention techniques including microstimulation (Schrimpf et al., 2024). Building on this foundation, we move from offline evaluation to prospective *model-in-the-loop* use: we optimize stimulation sites and stimuli *in-silico* and, to our knowledge for the first time, test these model predictions *in-vivo* to run visual cortex stimulation experiments.

**Visualization of visual representations.** Recent work links deep image synthesis to the neural code in high-level vision. Bashivan et al. (2019) used a discriminative deep network (AlexNet) in a generative way to synthesize images that maximally drive or selectively control V4 population responses, and Ponce et al. (2019) used closed-loop optimization in the latent space of deep generators (DeePSim, BigGAN) to evolve images that strongly excite IT neurons. Building on GANs, Dado et al. (2024) and Papale et al. (2024) learned linear mappings between neural activity and GAN latents to reconstruct and optimize category-selective stimuli, while Shahbazi et al. (2024) introduced "perceptograms": GAN-generated images that animals behaviorally confuse with the state of being optogenetically stimulated in IT. We follow a similar strategy, but - lacking monkey neural recordings during perturbations - visualize *model* IT representations: simulated microstimulation perturbs topographic model IT activity, which we then map into GAN and diffusion latent spaces to illustrate the predicted perceptual consequences of stimulation. Our approach may thus provide intuitions about model IT representations.

## 3 METHODS

We combine electrophysiological recordings, model-guided microstimulation, and primate behavioral testing in a 3-stage process that closes the loop between computational models and primate experiments (**Fig. 2**). The experimental setup involves 2 macaque monkeys performing a two-alternative forced choice (2AFC) visual recognition task, with Utah electrode arrays implanted in their inferior temporal cortex (for details, see **Appendix Sec. A.2.1**). The goal of the stimulation is to bias monkey behavior towards one of the two visual choice targets.

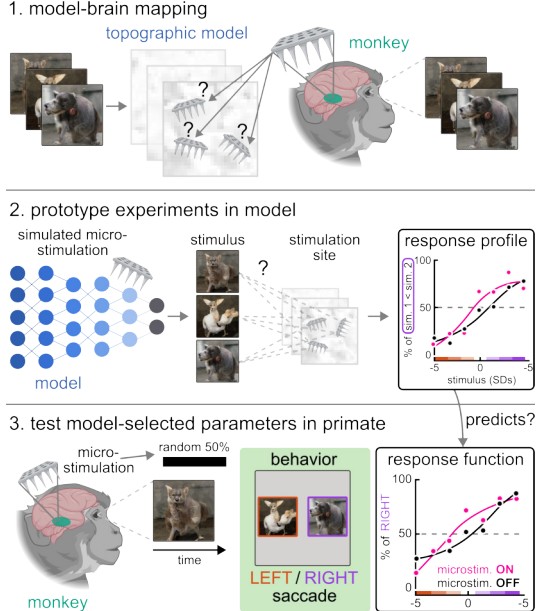

**Figure 2: Model-guided microstimulation. 1. Model–brain mapping.** To align model tissue and monkey brain recordings, we use passive-viewing responses of 4,000 images recorded 2–4 days before each experimental session. We then simulate various positionings of an electrode grid on the topographic tissue of model candidates, selecting the model grid position and orientation that maximizes correlations between model and monkey recording sites. This yields a fixed one-to-one mapping between sites in the model and brain-implanted electrode grid. **2. Prototype experiments in model.** For each candidate site we generate sequences of seven images varying smoothly along GAN latent space, rank them by a selectivity score (slope-to-noise), and test the effect of microstimulation on model-predicted 2AFC behavioral choices. Deepest-layer representations are converted to two-alternatives-forced-choice responses via similarity comparisons. **3. Test model-selected parameters in primate.** We select the top site–sequence predictions, mapping model neural sites back to the corresponding IT electrodes, and deploy the monkey experiment in a 2AFC recognition task. Biphasic trains of electric stimulation are delivered on designated trials, interleaved with sham. Full details in **Appendix Sec. A.2.1**.

To model this experiment, we first align topographic models to subject-specific passive-viewing electrophysiological recordings. Second, we prototype microstimulation experiments in the model to identify the most promising experimental setting. And third, we experimentally test whether the model-optimized combination of visual stimuli and stimulation sites yields predicted behavioral shifts when testing them in-vivo.

## 3.1 MODEL-BRAIN MAPPING

**Topographic models.** For modeling the effects of microstimulation and behavior we first train topographic deep artificial neural networks (TDANNs) based on the ResNet18 architecture (He et al., 2015) using an approach from Margalit et al. (2024) to incorporate spatial organization principles observed in biological vision. Before training the units of a given layer are assigned to a unique location on a 2D-plane that serves as the model cortical sheet. This spatial arrangement of model units allows us during training to compute a spatial loss (details below) and during inference to represent the neural changes induced by simulated microstimulation (Schrimpf et al., 2024).

We optimized TDANNs using a combined self-supervision ($L_{task}$; SimCLR, Chen et al. 2020a) and spatial loss (SL)

$$\text{TDANN Loss} \ = \ L_{task} \ + \ \sum_{k \,\in\, \text{layers}} \alpha_k \, SL_k, \qquad \alpha_k = 0.25.$$

For each layer $k$ and batch, we sample local cortical neighborhoods on the layer's 2D sheet and, within each sampled neighborhood, we sample unit pairs $(i, j)$. For each selected pair we compute (1) a response similarity $r_{ij}$ as the Pearson correlation between the units' activation vectors across stimuli, and (2) an inverse-distance weight $D_{ij} = 1/(d_{ij} + 1)$, where $d_{ij}$ is the Euclidean distance between their fixed cortical coordinates. By repeating this procedure for all sampled pairs, we obtain vectors $r$ and $D$. Following Margalit et al. (2024), we instantiate the spatial term as the relative spatial loss ($\text{SL}_k$):

$$\text{SL}_k = 1 - \text{Corr}(r, D),$$

which encourages nearby units to have more correlated responses, yielding smoothly varying maps across layers. For example, model early visual regions show orientation preference maps forming 'pinwheels' - where the preferred orientation of neurons rotates smoothly along all possible orientations from 0 to 180 degrees – that are known to exist in early visual areas across species (Kaschube et al., 2010). Additionally, model higher-level visual regions in the deepest layer show category-selective regions similar to higher-level visual cortex in humans and non-human primates (Kanwisher, 2017; Tsao et al., 2003; 2006; Freiwald et al., 2009).

We trained candidate topographic models on combinations of image datasets including ecoset (Mehrer et al., 2021), ImageNet (Russakovsky et al., 2015), Labeled Faces in the Wild (LFW, Huang et al. 2008), and VGGFaces2 (Cao et al. 2018; for details, see **Appendix Table 1**). For all model training, we used the same set of hyperparameters as Margalit et al. (2024): 200 epochs, initial learning rate: 0.6 (cosine decay), momentum: 0.9, batch size: 512. Weights are frozen after training such that the models' neural activity only depends on visual and stimulation input.

**Topographic mapping procedure.** Using passive-viewing responses of 4,000 images randomly sampled from a GAN latent space, recorded 2–4 days before each stimulation session, from Utah arrays in monkey inferior temporal cortex, we map neural sites in the model to neural sites in the brain implants. To do so, we first computed a linear predictivity score from the TDANN's deepest layer to each Utah array in IT. Concretely, for each model instance and each array, we extracted model activations to the 4,000 GAN images and fit a 10-fold cross-validated ridge regression. We quantified predictivity using the explained variance ($R^2$) across folds and retained those model–array pairs with the highest $R^2$ averaged across folds for subsequent topographic alignment. Importantly, linear predictivity ($R^2$) showed a large range across combinations of models and monkey arrays (monkey 1: [-0.06, 0.27], monkey 2: [0.05, 0.19]), indicating that the model and monkey array selection through this mapping procedure is indeed important. In a control analysis with an otherwise-matched non-topographic ResNet-18 trained without spatial loss ($\alpha_k = 0$), we observed comparable IT predictivity in monkey 1 (February session: $R^2 = 0.27$, June session: $R^2 = 0.19$ on the same IT array). This indicates that introducing topography does not substantially change the linear model–brain alignment in this dataset.

We then aligned simulated arrays and arrays used in-vivo by comparing their responses to a subset of the same 4,000 reference images. Specifically, we presented the images to both the monkey and the model, and for each candidate placement of a simulated Utah array on the model's cortical sheet we correlated responses site by site with the monkey array. We averaged these correlations across all 64 electrodes and selected the placement and orientation that yielded the highest overall match, thereby establishing a one-to-one correspondence between model and monkey electrodes.

## 3.2 PROTOTYPING EXPERIMENTS IN-SILICO

Our goal was to select, for each monkey, a specific electrode in inferior temporal cortex and a specific image sequence that together produce the strongest stimulation-induced behavioral outcome in a complex visual recognition task. To this end, we optimized experimental parameters in model space before mapping them back to the animal (**Fig. 2**). For more detailed implementation details, see **Appendix Sec. A.1**.

**Perturbation modules.** Recent evidence shows that topographic deep artificial neural networks can predict the behavioral outcomes of causal intervention techniques such as microstimulation (Schrimpf et al., 2024). We adapt this approach in stimulation modules that simulate the effects of microstimulation on nearby neural tissue. These modules operate by applying localized activity changes to model units, simulating the magnitude and spatial spread of electrical microstimulation as observed in experimental studies. We parameterized the perturbation modules based on em-

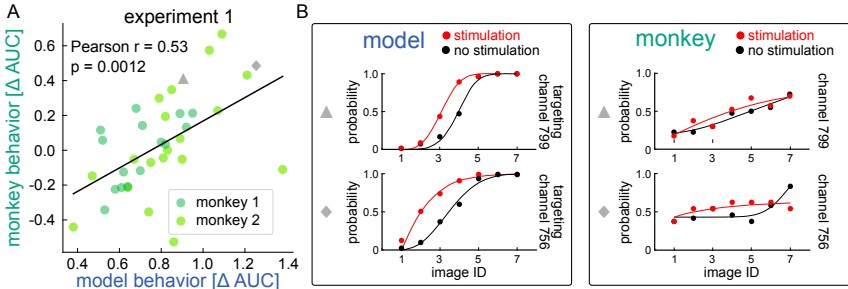

**Figure 3: Model predictions correlate with stimulation-evoked behavioral shifts. A)** Model-predicted behavioral shifts ($\Delta$AUC) correlate with stimulation-evoked shifts in the monkeys' behavioral responses ($\Delta$AUC), both when combining across the two subjects (Pearson $r = 0.53$, $p = 0.0012$). **B)** Example psychometric functions from two stimulation sites (gray symbols in A).

pirical data from prior microstimulation experiments in primate inferior temporal cortex (Stoney et al., 1968; Histed et al., 2009; Majaj et al., 2015; Kumaravelu et al., 2022). The magnitude of induced activity changes followed established current-distance relationships, with stimulation effects decreasing as a function of distance from the stimulation site.

Formally, following Schrimpf et al. (2024), the perturbation of a unit at cortical distance $d$ from the stimulation electrode is defined as

$$\Delta r(d) = \min\left(r_{\text{base}} + \gamma \cdot f_{\text{pulse}},\, r_{\text{max}}\right) \cdot \exp\left(-\frac{d}{\lambda(I)}\right), \tag{1}$$

where $r_{\text{base}}$ denotes the baseline firing rate (set to 30, Hz, consistent with primate IT recordings), $f_{\text{pulse}}$ is the stimulation pulse frequency (Hz), and $\gamma$ a gain factor linking pulse frequency to firing rate increase under a linear assumption. To prevent unrealistically high activity (Ponce et al., 2019), firing rates are clipped at $r_{\text{max}} = 200$Hz. The distance-dependent decay is captured by $d$, the cortical distance (mm) from the electrode, and $\lambda(I)$, a spatial decay constant (mm) that increases with stimulation current $I$ (µA), reflecting the broader spread of activity at higher currents. Thus, stimulation increases activity proportionally to the pulse rate at the electrode, saturates at a maximum firing rate, and falls off exponentially with cortical distance, consistent with current–distance relations from empirical studies.

**Stimulus generation.** We adopted the stimulus generation method of Papale et al. (2024) based on a StyleGAN-XL (Sauer et al., 2022), which links neural activity patterns in inferotemporal (IT) cortex to a generative adversarial network (GAN) latent space. Specifically, a linear mapping between multi-unit activity (MUA) from inferior temporal cortex recordings and the GAN's latent vectors (512 dimensions) was estimated from 4,000 reference images. This mapping enabled reconstruction of seen stimuli and, critically, of systematic perturbation of neural activity at individual cortical sites. By linearly adding or subtracting up to five standard deviations of the response at a targeted site to the 4,000 reference images, while keeping activity at other sites fixed, we generated naturalistic seven-image sequences in GAN image space. Each sequence thus corresponded to a parametric modulation of the targeted site's response, reflecting its neural tuning dimension. These GAN-derived image sequences then served as candidate stimuli for both in-silico and in-vivo microstimulation experiments, where they allowed us to test whether stimulation could bias neural activity and perceptual choices along the dimension to which the targeted site was tuned.

**Procedure.** For each candidate site, we (1) generate sequences of seven images to systematically modulate a site's activity; (2) rank sequences by a simple selectivity score (slope-to-noise), favoring monotonic, site-specific modulation, and (3) run in-silico perturbation experiments via the microstimulation module. To predict behavioral outcomes, we convert penultimate layer representations to 2AFC responses via similarity comparisons between the sample image and the two alternatives of a given trial (for details of the experimental design in model and monkey, see **Appendix Fig. 7**). To approximate trial variability in otherwise deterministic models, we aggregate across 30 top-ranked sequences per site to obtain smooth psychometric curves and summarize stimulation strength by $\Delta$AUC (perturbed − unperturbed trials).

### 3.3 TESTING MODEL PREDICTIONS IN-VIVO

The final step is to test model predictions experimentally. To do so, we select the top site–sequence pairs predicted by the model to evoke the highest behavioral change and map the neural sites back to monkey electrodes via the model-brain mapping established in step 1. Following Papale et al. (2024) monkeys performed a two-alternatives-forced-choice visual recognition task. During each trial we first present a target stimulus (one of the possible seven images of a sequence) followed by the simultaneous presentation of two alternative images (always two images at the two extremes of the given sequence). To indicate which of the two alternative images the animal perceived as more similar to the target image, they are trained to make a saccade to where the selected alternative image was presented. We applied biphasic microstimulation trains using chronically implanted Utah arrays (for details, see **Appendix Fig. 7**).

Stimulation and sham trials were randomly interleaved (50% / 50%) and choices were read out from the alternatives used in model prototyping (sequence extremes). We quantified stimulation-evoked shifts in choice probability between unperturbed and perturbed trials and report effects as $\Delta$AUC. Due to degrading signal quality (**Appendix Fig. 10**) the implants are now explanted from both animals, thus no additional experimental runs can be performed using our setup.

### 3.4 VISUALIZATIONS

We visualize the perceptual consequences of model-guided microstimulation using two complementary generative pipelines (see **Appendix A.3**). First, we adapt the Brain2GAN framework (Dado et al., 2024) to our setting by replacing neural recordings with the deepest-layer activations of a topographic vision model, and by pairing these activations with images generated by a pretrained StyleGAN-XL (Sauer et al., 2022) in its feature-disentangled $\mathbf{w}$-space. We learn a ridge regression on topographic activations to $\mathbf{w}$-latents using only unperturbed activity, and then synthesize images from model states with and without simulated microstimulation. Because the GAN and ridge regression are fixed, differences between unperturbed and perturbed reconstructions directly reflect the changes in the model's internal representation induced by stimulation at specific cortical locations and simulated current levels.

Second, we use a diffusion-based pipeline built on Stable Diffusion v1.5 with IP-Adapter (Rombach et al., 2022; Ye et al., 2023), which conditions the denoising process on both a text prompt and a CLIP (Radford et al., 2021) vision embedding. We generate a train/test image set from prompts based on the ecoset category structure (Mehrer et al., 2021), and fit a ridge regression from deepest-layer topographic features to CLIP vision embeddings. At test time, we hold the text prompt, noise latent, and diffusion hyperparameters fixed and vary only the simulated microstimulation in the model, decoding the resulting perturbed activations into CLIP space and re-running the diffusion process. In both the GAN and diffusion pipelines, we additionally perform a shuffled-perturbation control: the targeted perturbation at a maximally face-selective site is randomly permuted across feature dimensions before decoding. This control preserves the overall perturbation magnitude but destroys its topographic structure, allowing us to test whether the emergence or amplification of face-like content in the generated images specifically depends on spatially structured stimulation of face-selective regions rather than on non-specific global modulation.

## 4 RESULTS

We evaluated two complementary measures of stimulation effectiveness. First, we asked whether the magnitude of stimulation-evoked behavioral shifts in the monkey was predicted by the magnitude of shifts in our model (model–monkey Pearson-r correlation). Second, we asked whether stimulation in the monkey induced a consistent behavioral shift away from baseline (monkey $\Delta$AUC significantly greater than zero).

We performed two experiments with the same experimental setup that differ slightly in the way we pre-selected the monkey stimulation sites (**Appendix Sec. A.1**). Electrode implants were relatively stable for monkey 1 where we could perform both experiments, but the absolute number of tested sites still decreased from experiment 1 (13) to experiment 2 (9 sites). Decreasing neural signal quality (**Appendix Fig. 10**) only allowed us to perform one experiment in monkey 2. The two experiments only differ in the spatial constraint applied to select candidate electrodes of

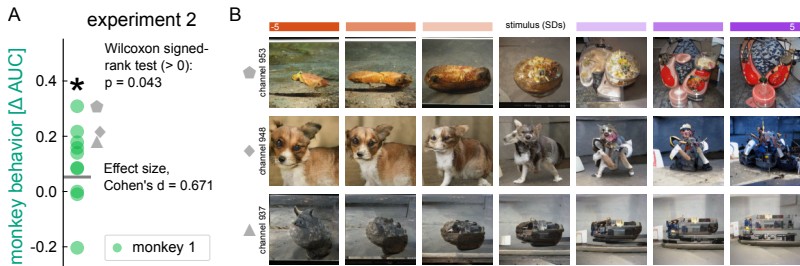

**Figure 4: Model-guided stimulation biases monkey behavior. A)** In experiment 2, model-guided stimulation induced a significant behavioral shift in monkey 1 (Wilcoxon signed rank test: $p = 0.043$; Cohen's $d = 0.67$). Due to declining signal quality, experiment 2 could not be conducted with monkey 2. **B)** Three example GAN-generated image sequences used for stimulation (images 1, 4, and 7 shown from each seven-image sequence; corresponding sites highlighted in A).

a given Utah-array. Specifically, in experiment 1 we used a Manhattan distance between candidate stimulation sites of 1.6mm, whereas we reduced this limit to 1.2mm in experiment 2 to allow for a larger number of candidate stimulation sites. In other words, the spatial constraint was loosened in experiment 2 to allow for a higher percentage of stimulation sites to be tested.

### 4.1    PREDICTING STIMULATION-EVOKED BEHAVIORAL SHIFTS (EXPERIMENT 1)

We found that model-predicted behavioral shifts were positively associated with stimulation-evoked shifts measured in both monkeys in experiment 1 (**Fig. 3**). Specifically, the model predictions (in $\Delta$AUC, x-axis) versus monkey behavior (in $\Delta$AUC, y-axis) revealed robust correlations in both animals (monkey 1: Pearson $r = 0.58$, $p = 0.024$, $r^2 = 0.34$, monkey 2: Pearson $r = 0.53$, $p = 0.019$, $r^2 = 0.28$). Bootstrap resampling yielded confidence intervals confirming that the estimated effect is reliably above zero in both animals while their range reflects the small sample size (monkey 1: 95% CI: $r \in [0.20, 0.89]$, $R^2 \in [0.04, 0.79]$; monkey 2: 95% CI: $r \in [0.16, 0.85]$, $R^2 \in [0.03, 0.73]$). To quantify how much of these effects could arise by chance, we shuffled behavioral labels and recomputed $R^2$ 1,000 times. This yielded null distribution with medians close to 0 (monkey 1: median $R^2 = 0.034$; 95% CI $[6.8 \times 10^{-5}, 0.32]$; monkey 2: median $R^2 = 0.029$; 95% CI $[1.3 \times 10^{-4}, 0.24]$). For both animals, the observed $R^2$ exceeded nearly all null samples (monkey 1: permutation $p = 0.019$; monkey 2: $p = 0.017$) This indicates that combinations of stimulation site and GAN image sequence predicted by our model to yield stronger behavioral effects, tended to have a stronger behavioral effect in the monkey. However, the monkey behavioral responses were not significantly greater than zero (Wilcoxon signed rank test, $p > 0.05$) in this first experiment.

### 4.2    INDUCING BEHAVIORAL BIAS ALONG A TARGETED DIRECTION (EXPERIMENT 2)

Due to degrading signal quality of the implanted neural recording devices in monkey 2 (**Appendix Fig. 10**), we were only able to perform experiment 2 with more candidate stimulation sites in monkey 1. In this experiment, monkey behavioral responses were significantly shifted above a null baseline (Wilcoxon signed-rank test; $p = 0.043$, effect size Cohen's $d = 0.671$), indicating that parameters predicted by the model indeed yielded a reliable behavioral effect in-vivo (**Fig. 4**). However, the larger number of candidate stimulation sites available in this experimental setting no longer yielded evidence for per-site and per-image-sequence predictive behavioral power of our model ($p > 0.05$). We believe this reduced effect is mainly due to degraded signal quality (**Appendix Fig. 10**).

For both experiment 1 and 2 we cannot fully exclude contributions from global state changes, but several aspects of our design make explanations mainly based on attention/arousal effects unlikely. Stimulation and non-stimulation trials were randomly interleaved at 50% / 50%, minimizing expectation-driven state changes. What is more, we are not aware of evidence that inferior temporal microstimulation induces non-specific attentional or arousal effects that would be able to steer behavior in a targeted way.

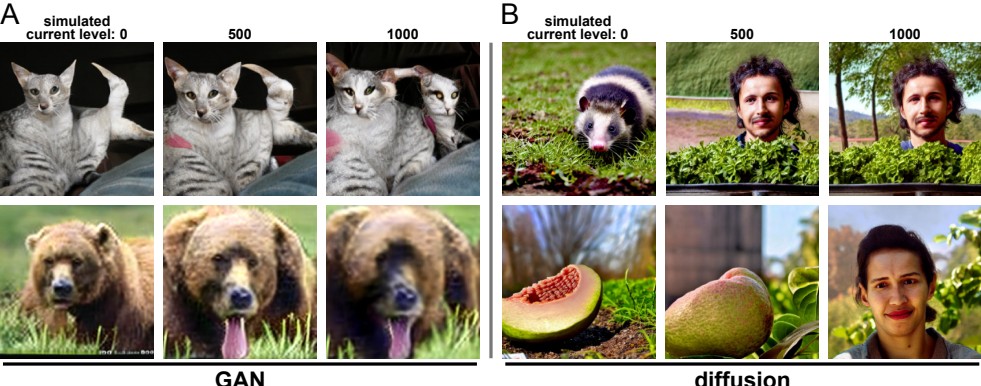

**Figure 5: Visualizing perceptual effects of microstimulation.** We visualize the effects of stimulation using a GAN-based approach (A), and a diffusion-based approach (B) and show two examples for each. Simulated current amplitude increases from left ($0\,\mu$A) to right ($1000\,\mu$A) in both panels. A) In the first row, stimulation transforms a cat's tail into an additional face, while in the second row it enlarges the face of a bear. B) The first row shows how stimulation transforms a bug-like creature into a face, while in the second row a fruit is transformed into a human face.

### 4.3 VISUALIZATION OF IN-SILICO PERCEPTUAL EFFECTS OF PERTURBATION

Beyond shifts in behavioral choices, a key question is how neural stimulation alters the perceptual content of visual experience. For instance, Schalk et al. (2017) examined a 26-year-old patient with intractable epilepsy who was implanted with 188 subdural electrodes also covering higher-level visual cortex to localize seizure foci. During stimulation, electrodes over a face-selective cortical region evoked illusory faces ("facephenes") that appeared superimposed on any object the patient was viewing, whereas stimulation of color-selective sites produced illusory "rainbows". These observations provide causal evidence, based on a subjective report, that stimulation of category-selective cortex can induce highly specific perceptual changes.

Inspired by this approach, we aimed to visualize the perceptual consequences of stimulation in our model-based framework. Papale et al. (2024) recently introduced a method for reconstructing perceived images by projecting neural activity patterns into the latent space of a generative adversarial network (GAN). Here we adapted this technique to examine the effects of perturbations in face-selective regions of our topographic model.

**Mapping model states to image space.** We trained a linear mapping from the deepest layer of the topographic model (model inferior temporal cortex, 25,088 units) into the latent space of the GAN used for stimulus generation (512 dimensions). This mapping was calibrated on 30,000 GAN-generated images, enabling us to project topographic model activity states into image space (for details, see **Appendix Fig. 9**). Reconstructions of unperturbed responses (simulated stimulation current$= 0\mu$A) closely matched the original stimuli (ground truth), confirming that the mapping preserved key visual content (see two leftmost columns 'ground truth' vs. 'simulated stimulation current $= 0\mu$A' in **Appendix Figs. 12,13,15**).

**Qualitative stimulation effects.** We then perturbed model sites with high face-selectivity as defined by a functional face localizer from neuroscience (Stigliani et al., 2015) while presenting objects in an independent set of 5,000 images not used to establish the linear mapping between the topographic model and the GAN. In several cases, the reconstructions revealed face-like features superimposed on the original object, resembling the "facephenes" reported in human stimulation studies (Schalk et al., 2017). For example, stimulating a site at the center of a face-selective region in model inferior temporal cortex with a simulated current level of $1000\,\mu$A added an illusory second face to an image of a cat or increased the area of an image occupied by a bear's face (**Fig. 5**). Similar effects were observed for other objects, whereas stimulation at control sites with low face-selectivity did not reliably introduce such face-like structure. To provide an overview, we present exhaustive combinations of stimulation site coordinates corresponding to varying levels of face-selectivity of the underlying simulated cortical sheet and simulated current levels in the appendix (**Appendix Figs. 12,13**).

**Interpretation.** These visualizations provide an interpretable window into the otherwise inaccessible perceptual consequences of microstimulation. They suggest that activating face-selective regions can bias representations of unrelated objects toward the preferred category, echoing the patient reports of "facephenes".

## 5 DISCUSSION

We introduce a model-guided framework that links topographic deep networks, in-silico perturbations, and an explicit model-to-monkey electrode mapping to steer primate visual behavior via microstimulation in inferior temporal (IT) cortex. Across two animals, model-derived combinations of stimulation site and image sequence yield positive correlations between model and monkey behavior in experiment 1, and lead to a stimulation-driven in-vivo behavioral shift in experiment 2. Together, these findings establish a proof-of-principle: topographic models with perturbation modules can guide causal interventions that bias in-vivo behavior in response to complex visual objects.

**Limitations.** The main limitation of this study is the degrading signal that prevents more thorough testing to support our claim that model-guidance can steer primate behavior. With a stable signal quality we could have performed additional experiments in both animals to test whether further improvements on our model-guided microstimulation framework result in larger effect sizes than those we describe. Without additional experiments our results are split between two experimental sessions, where either the monkey behavioral effect is not significantly different from zero (experiment 1), or where there is no clear correlation between model and monkey behavior (experiment 2).

**Next steps.** Additional experiments would allow for further baselines, e.g. testing whether a random selection of both stimulation site and image sequence does indeed not result in the same behavioral changes we observed. Future experiments could further investigate alternative topographic models (Lu et al., 2023; Deb et al., 2025), including the relevance of topography in the first place, how to best perform model-brain mapping, and details of the perturbation module (additive vs. multiplicative modulation, optimal current level, single- vs. multi-site stimulation).

**Toward clinical impact.** By shifting the target from early retinotopic codes to higher-level object codes and using models to plan interventions, our framework outlines a computational backbone for next-generation visual prosthetics aimed at restoring percepts of complex visual objects. In other words, we believe it is possible that IT representations are not only necessary, but sufficient to represent complex visual objects such as faces and scenes. If so, the effect of inducing activity patterns in IT would not be limited to changes at a categorical level, but could capture a more fine-grain level allowing for e.g. the modulation of single face-features by stimulating face-feature-selective regions (Issa & DiCarlo, 2012). Whether this holds true or whether simultaneous stimulation of earlier visual areas is required to elicit percepts of complex visual objects on a detailed level, is, in our eyes, in our eyes an empirical question. More broadly, model-guided stimulation may be applicable beyond vision – for example, selecting input stimuli and stimulation patterns for a range of causal intervention techniques such as microstimulation, but also transcranial magnetic stimulation, or focused ultrasound to diagnose and treat neuropsychiatric disorders.

## 6 CONCLUSION

By aligning topographic models to higher-level visual cortex, optimizing stimulation sites and stimuli *in-silico*, and testing the model-predicted experimental parameters *in-vivo*, we establish a practical model-in-the-loop framework for guiding causal interventions in vision. Model predictions were associated with stimulation-evoked shifts in behavioral responses, and with a bias along a targeted perceptual dimension. Image reconstructions from perturbed model activity further illustrated perceptual consequences, with stimulation at face-selective sites biasing representations toward faces. Together, these results define a pipeline in which topographic models with perturbation modules inform experimental design and predict behavioral outcomes. Our approach might extend to more advanced protocols such as multi-site stimulation, and to other causal intervention techniques. By targeting higher-level visual regions rather than early visual cortex, our framework offers a computational foundation for prosthetic strategies aimed at eliciting richer, object-level visual experiences and supports closed-loop optimization with translational promise.

ACKNOWLEDGMENTS

We thank Pieter Roelfsema and the EPFL NeuroAI lab for useful discussions. We are grateful for funding from Schmidt Sciences (G-25-69784), SNF (10.003.772), SNF Spark (CRSK-3_228579), and ETH Domain Open Research Data (5th and 6th call).

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

# A APPENDIX

| ImageNet | ecoset | LFW | VGGFaces2 | # of instances |
|:---:|:---:|:---:|:---:|:---:|
| ✓ | | | | 5 |
| ✓ | | ✓ | | 10 |
| ✓ | | | ✓ | 3 |
| | ✓ | | | 10 |
| | ✓ | ✓ | | 10 |
| ✓ | ✓ | | | 5 |

**Table 1: Training data of candidate models.** Before model selection (see Sec. A.1) models are trained on different combinations of image sets. We refer to two models with the same architecture trained with the same hyperparameters and on the same sets of images and only differing in their set of initial weights as two model instances.

## A.1 OPTIMIZING MICROSTIMULATION IN MONKEYS THROUGH SIMULATIONS OF EXPERIMENTS IN MODELS

**Passive viewing data recording.** Two days prior to stimulation experiments we conducted passive viewing sessions to calibrate the models. Each monkey passively viewed 4,000 reference images randomly sampled from GAN latent space while we recorded activity from all available IT electrodes in each animal.

**Stimulation site pre-selection in monkey array.** We selected electrodes in monkey arrays based on signal quality (split-half reliability across 4,000 reference images) and a spatial constraint avoiding tissue damage (minimum spacing of 1.6mm (experiment 1) or 1.2mm (experiment 2) Manhattan distance between any two electrodes used in experiments).

**Selection of model, and of monkey array.** For each model instance, we computed cross-validated linear predictivity from model inferior temporal cortex to monkey arrays using the 4,000 reference images. We then selected the best combinations of monkey array and model with regard to the expected behavioral outcome.

**Stimulus generation.** We used a generative-adversarial-network-based approach pioneered by Papale et al. 2024 to generate sequences of 7 images optimized to modulate neural activation level at a targeted stimulation site in monkey inferior temporal cortex.

**Placing a simulated Utah array on the model cortical sheet.** We identified the simulated Utah array location and orientation on the model equivalent of inferior temporal cortex that best correlate with a monkey array of interest using a subset of the responses to the 4000 reference images.

**Ranking image sequences by a selectivity index.** We sorted stimulation site-specific image sequences by a selectivity index reflecting its ability to modulate neural activation levels at the targeted site in a monotonically increasing or decreasing way. By favoring monotonic changes, we attempt to bring the neural changes resulting from visual stimulation as close as possible to the neural changes resulting from the stimulation.

**In-silico perturbation experiment and behavioral readout.** We performed the experiment in the model using the selected combination of image sequence and stimulation site in model IT. We considered the model behavioral response ($\Delta$AUC of psychometric curves: perturbed $-$ unperturbed trials) as the prediction of the monkey behavioral response.

**Mapping from model to monkey stimulation device.** We projected model stimulation sites yielding strongest behavioral model responses to the monkey cortex using the 1:1 mapping between simulated model and monkey electrodes used for placing the simulated Utah array (see above).

**Model-guided microstimulation in monkeys.** Monkeys performed the delayed-match-to-sample-task using the model-selected stimulation sites and image sequences.

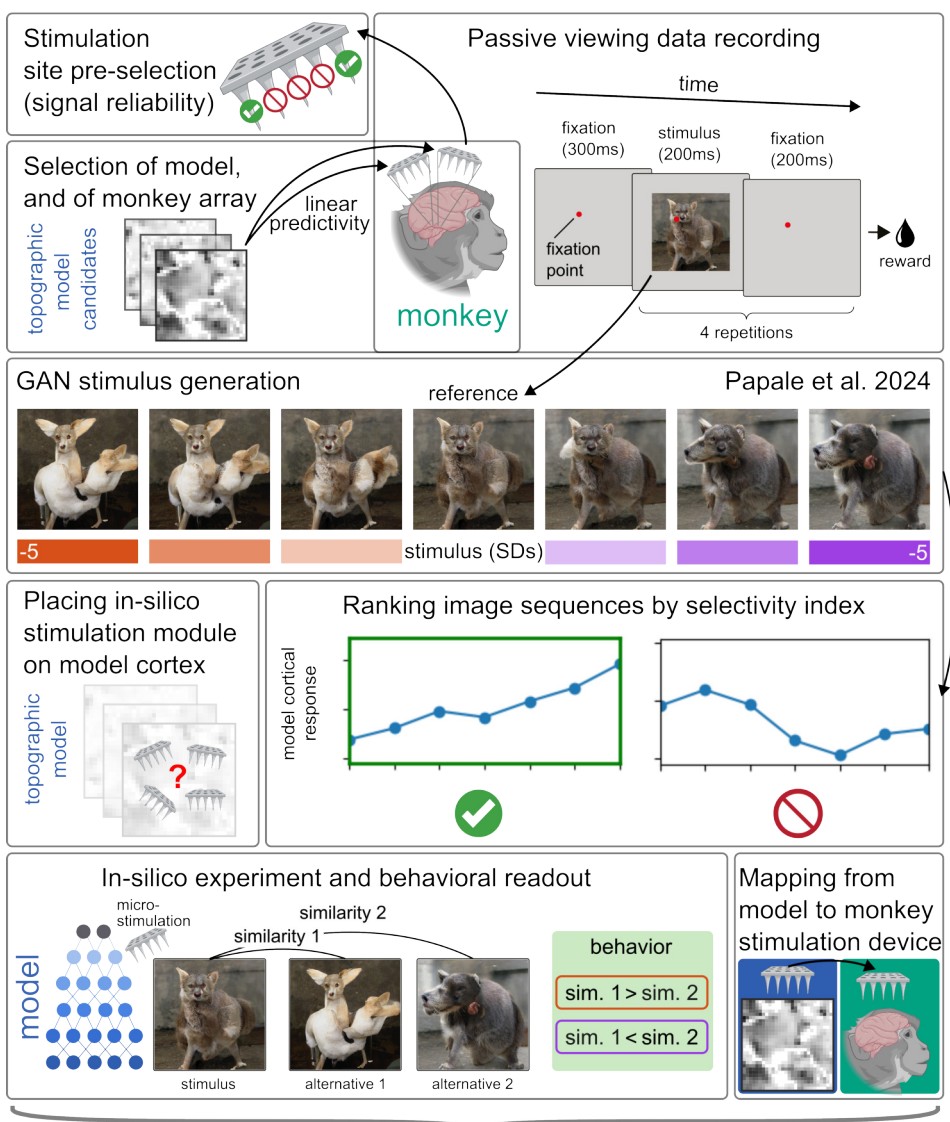

**Figure 6: Simulating and optimizing monkey behavioral responses via topographic models** For details, please see list in main text.

## A.2 EXPERIMENTAL PARADIGM

### A.2.1 MONKEY SETUP

The experimental design follows the experimental paradigm established by Papale et al. (2024), with modifications to integrate model-guidance for stimulation site and stimulus selection (**Fig. 7A**). For all information on animal care and housing, surgeries for implanting stimulation devices, electrophysiology including multi-unit-activity pre-processing, intracortical microstimulation, stimulus presentation, and ethical approval for animal testing we refer to (Papale et al., 2024).

Two male macaque monkeys were implanted with Utah arrays in posterior inferior temporal cortex and trained to perform a delayed match-to-sample task. On each trial, the monkey fixated a central point before a sample stimulus was presented for 200ms, followed by a 100ms gray screen. The sample stimulus was one of a sequence of 7 images optimized to drive the response profile of the stimulation site in the direction of the vector induced by stimulation. After this delay of 100ms, two

choice images appeared on opposite sides of the screen, corresponding to the extremes of the GAN-generated image sequence. After 400ms, the fixation point disappeared, cueing the animal to make a saccade to the image judged most similar to the sample stimulus presented before. Correct responses were rewarded with juice. When the central reference image of the associated GAN sequence of 7 images serves as the sample stimulus, rewards are delivered randomly on 50% of trials to avoid bias in either direction.

To assess the influence of microstimulation on visual behavior, electrical microstimulation was applied in 50% of trials via one IT electrode (200ms, 50μA, 300Hz biphasic pulses) during and shortly after the sample presentation window (75–275ms after stimulus onset). Trials without stimulation served as a baseline conditions (sham), allowing direct comparison of behavioral choices with and without stimulation.

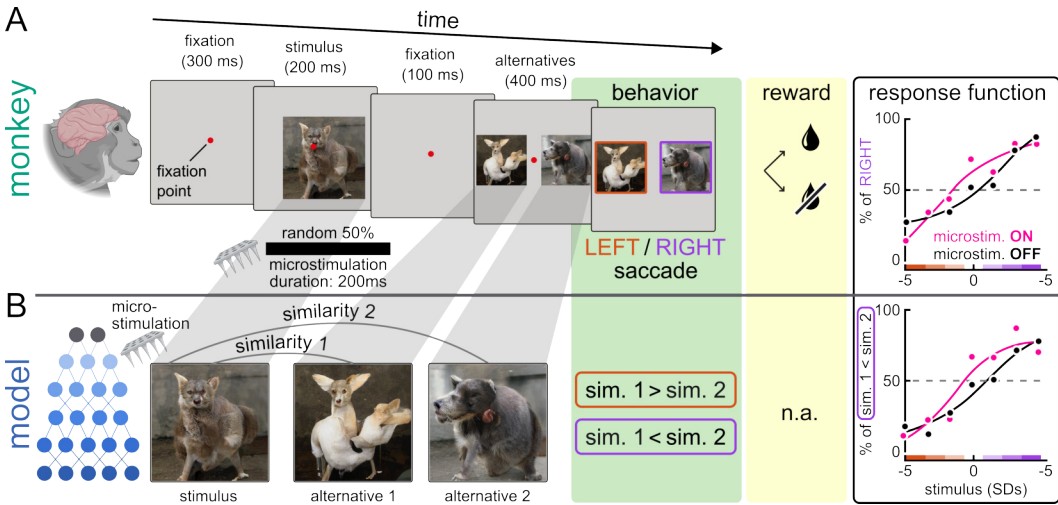

**Figure 7: Experimental setup in monkey and model. A)** Monkeys were trained to perform a visual delayed match-to-sample-task. The animals had to select which of two alternative images they perceive as more similar to a single stimulus shown at the beginning of a trial. Correct responses were rewarded with juice. We computed a psychometric function based on the responses across multiple stimuli. Monkey experimental design including stimulus generation as in (Papale et al., 2024). **B)** Model simulation of monkey experiment. We used topographic model to simulate the experimental design shown in A as follows. We extracted model activations in response to the two alternative images and the sample stimulus. If the similarity between alternative 1 and the sample stimulus was higher than the similarity between alternative 2 and the sample stimulus, this was scored as a behavioral response towards alternative 1 and vice versa. The psychometric response function across multiple stimuli was computed in the same way as in the monkey experiment.

### A.2.2 Topographic model setup

To simulate the perturbation experiment in-silico, we employed topographic vision models with an explicit perturbation module (**Fig. 7B**). As there is no notion of time in our feed-forward models, a trial was simulated by presenting the stimulus of interest — one of the seven images constituting a GAN-derived sequence — together with the first and the last image of that sequence. We then extracted the activations from the deepest layer and computed the similarity between the stimulus representation and each of the two extremes. The model's behavioral choice was defined by the extreme with higher similarity: if the stimulus was more similar to the last image than to the first, the model was scored as having chosen the last image, whereas if the reverse was true it was scored as having chosen the first.

Trials without activating the perturbation module provided the baseline condition, corresponding to trials without microstimulation in the monkey experiment. For stimulation trials, we applied a local perturbation to the model units corresponding to the targeted cortical site during stimulus presentation, thereby simulating the effect of electrical microstimulation. We compared the resulting

choice probabilities between baseline and stimulation conditions following the same analysis as in the monkey experimental setup.

A key difference to in-vivo experiments is how we treat variability across trials in our models. In the monkey experiment, each trial is a repeated event in time: the same stimulus can yield different choices due to biological variability such as varying levels of attention, fatigue, or other forms of noise. In contrast, our models are fully deterministic: the same input image and perturbation always produce the same output. As a consequence, the psychometric function derived from a single image sequence is binary in shape consisting entirely of zeros, entirely of ones, or describes a sharp step function rather than the smooth, graded functions typically observed in behavior or biological organisms. To approximate biological variability and thus mimic trials, we therefore consider multiple GAN-generated sequences that are all optimized to modulate activity at the same monkey stimulation site. Each distinct sequence constitutes an in-silico trial, and averaging across these sequences provides a model analogue of the across-trial variability observed in the animal. We introduce variability on purpose on the data side so that any future model can be evaluated on the same image sequences without requiring model-specific noise mechanisms. If variability would be introduced by e.g. dropout or noise directly applied to weights or activations, each model would require its own implementation for it to be tested and compared on our data.

## A.3 VISUALIZATION OF PERCEPTUAL EFFECTS OF STIMULATION

We use two complementary generative pipelines to visualize the perceptual consequences of simulated microstimulation in the topographic model: a GAN-based reconstruction approach and a diffusion-based approach based on Stable Diffusion with IP-Adapter. In both cases, the core idea is to learn a linear decoder from topographic model activity into the conditioning space of a pretrained generator, and then compare images synthesized from unperturbed versus perturbed model states.

### A.3.1 GAN-BASED VISUALIZATIONS

Our GAN-based visualization procedure closely follows the Brain2GAN framework (Dado et al., 2024), with the primary difference being the source of the encoding features: we use the deepest layer of a topographic vision model as the encoder, whereas Brain2GAN uses neural recordings. As in Brain2GAN, we work in the feature-disentangled $\mathbf{w}$-latent space of a pretrained StyleGAN-XL generator (Sauer et al., 2022), rather than its original $\mathbf{z}$-space.

**Dataset construction.** We first generated a synthetic image dataset using a pretrained StyleGAN-XL generator. We sampled latent codes $\mathbf{z}$ from the StyleGAN-XL prior and mapped them through the StyleGAN-XL mapping network to obtain corresponding $\mathbf{w}$-latents. For each sampled latent, we an image and recorded:

- the StyleGAN-XL $\mathbf{w}$-latent $\mathbf{w}_i \in \mathbb{R}^{d_w}$, and
- the deepest-layer activation vector of the topographic vision encoder $f_i \in \mathbb{R}^{d_f}$,

where $d_w = 512$ and $d_f = 25{,}088$. In total, we generated 30,000 training images and 6,000 test images. All decoders are trained exclusively on unperturbed encoder activations.

**Linear mapping from topographic model to $\mathbf{w}$-space.** To map topographic model states into the StyleGAN-XL latent space, we fit a ridge regression from deepest-layer activations to $\mathbf{w}$-latents. For a training set $\{(f_i, \mathbf{w}_i)\}_{i=1}^{N}$, we learn a weight matrix $W \in \mathbb{R}^{d_w \times d_f}$ by minimizing

$$\mathcal{L}(W) = \sum_{i=1}^{N} \left\| \mathbf{w}_i - W f_i \right\|_2^2 + \alpha \left\| W \right\|_F^2, \tag{2}$$

where $\alpha > 0$ is the ridge regularization parameter and $\| \cdot \|_F$ denotes the Frobenius norm.

The regularization parameter $\alpha$ is selected via leave-one-out cross-validation on the 30,000-image training set. After selecting $\alpha$, we keep this linear map fixed for all subsequent visualizations.

Given a new encoder state $f$, the corresponding predicted StyleGAN-XL $\mathbf{w}$-latent is

$$\hat{\mathbf{w}} = W f. \tag{3}$$

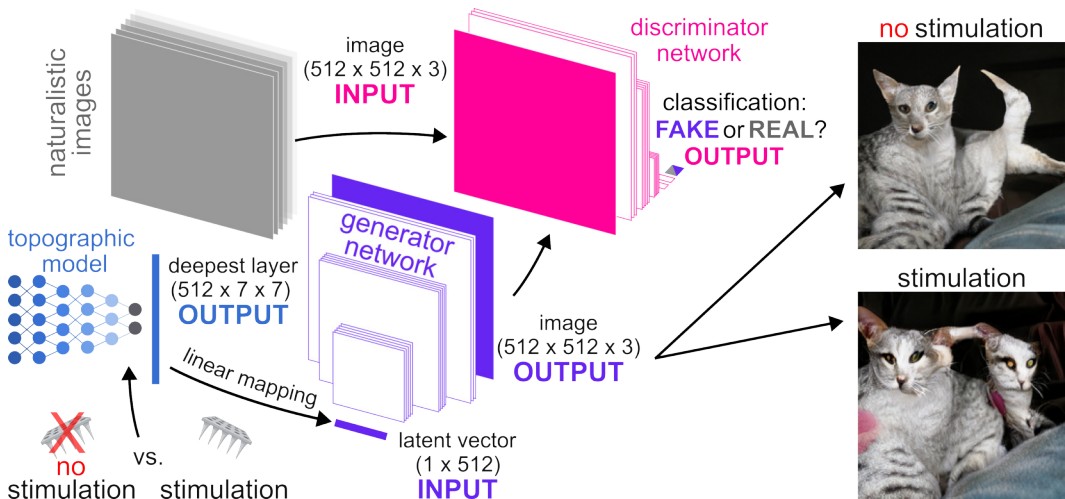

**Figure 8: GAN-based visualization procedure.** Neural activity patterns from the deepest layer of the topographic model are linearly mapped into the latent space of a generative adversarial network (GAN). This mapping enables reconstruction of unperturbed responses as well as visualization of the perceptual consequences of simulated microstimulation by comparing GAN reconstructions with and without perturbation.

**Baseline reconstructions.** To assess the quality of the decoder, we reconstruct test images from unperturbed encoder activations. For each test image $x$ with deepest-layer feature vector $f_{\text{unpert}}(x)$, we compute

$$\hat{\mathbf{w}}_{\text{unpert}}(x) = W f_{\text{unpert}}(x), \tag{4}$$

and pass $\hat{\mathbf{w}}_{\text{unpert}}(x)$ through the StyleGAN-XL synthesis network to obtain a reconstructed image $\hat{x}_{\text{unpert}}$. These reconstructions (current level $0\,\mu\text{A}$) serve as the baseline for the perturbation visualizations shown in the appendix.

**Perturbation conditions.** To visualize the perceptual effects of microstimulation, we apply the perturbation module to the topographic model at different stimulation sites and current levels. We focus on positions along the $y$-axis of the deepest-layer cortical sheet, with

- $y = 42$ corresponding to a highly face-selective region,
- $y = 20$ corresponding to a region with low face-selectivity,

and intermediate values (e.g. $y = 28, 34$) interpolating between these extremes. For each chosen site $(y)$ and simulated current level $I \in \{0, 100, 500, 1000\}\,\mu\text{A}$, we obtain a perturbed deepest-layer activation

$$f_{\text{pert}}(x; I, y) \tag{5}$$

by applying the perturbation module to the unperturbed state $f_{\text{unpert}}(x)$ for image $x$.

**GAN-based visualizations.** For each test image $x$, stimulation site $y$, and current level $I$, we compute

$$\hat{\mathbf{w}}_{\text{pert}}(x; I, y) = W f_{\text{pert}}(x; I, y), \tag{6}$$

and synthesize the corresponding image via StyleGAN-XL. Because the generator and decoder are fixed, any systematic differences between $\hat{x}_{\text{unpert}}$ and $\hat{x}_{\text{pert}}$ can be attributed to the changes in topographic model state induced by simulated microstimulation, rather than to changes in the generative model itself.

**Control perturbation ablation.** As a control, we test whether the observed effects depend on the spatial specificity of the perturbation pattern in the topographic model. For each current level $I$, we first compute a targeted perturbation at the maximally face-selective location $y_{\text{face}} = 42$. We then construct a "shuffled" perturbation by randomly permuting the entries of this perturbation vector across feature dimensions. This preserves the marginal distribution and overall magnitude of the

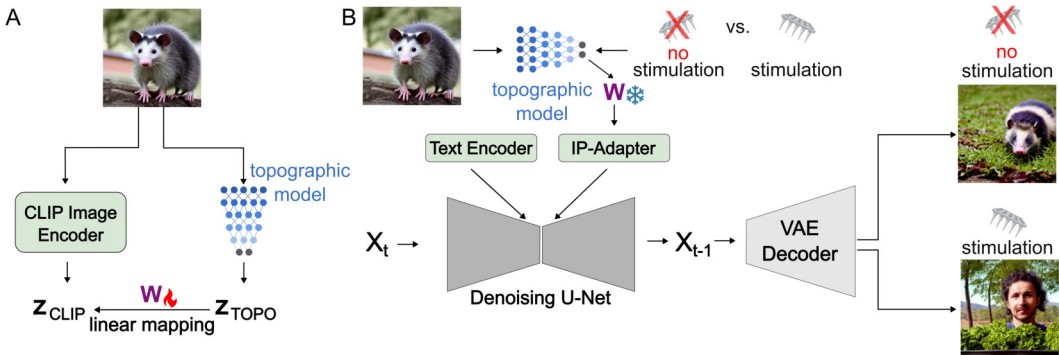

**Figure 9: Diffusion-based visualizations procedure. A)** Neural activity patterns from the deepest layer of the topographic model are linearly mapped into the latent space of CLIP. **B)** This linear mapping allows for transforming changes in TDANN activity from simulated microstimulation into changes in CLIP latent space that we use to condition the diffusion-based image synthesis via pretrained IP-Adapter weights.

perturbation but destroys its topographic structure. The shuffled perturbation is then applied to the unperturbed deepest-layer features and visualized through the same StyleGAN-XL generator. Comparing targeted and shuffled perturbations allows us to dissociate effects driven by feature-selective, spatially localized stimulation from those that could arise from non-specific global modulation with matched statistics.

### A.3.2 DIFFUSION-BASED VISUALIZATIONS WITH IP-ADAPTER

To complement the GAN-based reconstructions, we also visualize perturbation effects using a text-to-image diffusion model. Specifically, we use Stable Diffusion v1.5 with IP-Adapter (Rombach et al., 2022; Ye et al., 2023), which conditions the diffusion process on a CLIP Radford et al. (2021) vision embedding of an image in addition to a text prompt.

**Text prompts and generated dataset.** We construct text prompts by sampling ecoset category labels (Mehrer et al., 2021) and enriching them with quality and style modifiers, for example:

"a high-quality photo of a `[category]`, studio lighting, sharp focus"

For each prompt, we run Stable Diffusion v1.5 with IP-Adapter to generate images. During generation we record, for each image:

- the text prompt $p_i$,
- the initial noise latent $\epsilon_i$ that seeds the diffusion process,
- the resulting synthesized image $x_i$.

As before, we split the resulting dataset into training and test sets.

**Linear mapping from topographic model to CLIP vision space.** To couple the topographic model to the diffusion pipeline, we fit a ridge regression from deepest-layer activations to CLIP vision embeddings. For a training set $\{(f_i, \mathbf{c}_i)\}_{i=1}^N$ we learn a weight matrix $W_{\mathrm{CLIP}} \in \mathbb{R}^{d_c \times d_f}$ by minimizing

$$\mathcal{L}(W_{\mathrm{CLIP}}) \; = \; \sum_{i=1}^{N} \left\| \mathbf{c}_i - W_{\mathrm{CLIP}} f_i \right\|_2^2 \; + \; \alpha_{\mathrm{CLIP}} \left\| W_{\mathrm{CLIP}} \right\|_F^2, \tag{7}$$

where $\alpha_{\mathrm{CLIP}} > 0$ is selected via cross-validation on the training set. After training, we obtain predicted CLIP embeddings from topographic states via

$$\hat{\mathbf{c}} = W_{\mathrm{CLIP}} f. \tag{8}$$

At inference time, $\hat{\mathbf{c}}$ is used as a drop-in replacement for the CLIP vision encoder output in IP-Adapter.

**Controlled diffusion runs.** To visualize perturbation effects for a given test image $x$ and its associated text prompt $p$ and noise latent $\epsilon$, we proceed in two stages.

**Baseline (unperturbed) condition.**

1. Extract the deepest-layer activation $f_{\text{unpert}}(x)$ from the topographic model.
2. Compute the corresponding CLIP embedding

$$\hat{\mathbf{c}}_{\text{unpert}}(x) = W_{\text{CLIP}} f_{\text{unpert}}(x). \tag{9}$$

3. Run Stable Diffusion v1.5 with IP-Adapter using:
   - the stored text prompt $p$,
   - the stored noise latent $\epsilon$, and
   - $\hat{\mathbf{c}}_{\text{unpert}}(x)$ as the image-conditioning input.

   This yields a baseline image $\tilde{x}_{\text{unpert}}$ that approximates the original generated image under our topographic decoder.

**Perturbed condition.**

1. Apply the perturbation module at a chosen stimulation site $y$ and current level $I$ to obtain $f_{\text{pert}}(x; I, y)$.
2. Compute the perturbed CLIP embedding

$$\hat{\mathbf{c}}_{\text{pert}}(x; I, y) = W_{\text{CLIP}} f_{\text{pert}}(x; I, y). \tag{10}$$

3. Run Stable Diffusion with IP-Adapter again, using exactly the same text prompt $p$ and noise latent $\epsilon$, but replacing the image-conditioning embedding with $\hat{\mathbf{c}}_{\text{pert}}(x; I, y)$.

Because the text prompt, noise latent, and diffusion hyperparameters are held fixed between the unperturbed and perturbed runs, any systematic differences between $\tilde{x}_{\text{unpert}}$ and the perturbed image $\tilde{x}_{\text{pert}}(I, y)$ can be attributed to changes in the topographic model state induced by simulated microstimulation. As in the GAN-based case, we visualize these effects by varying both the stimulation site (e.g., along the face-selective axis of model IT) and the simulated current level, yielding image series that illustrate how perturbations bias the generative model's outputs.

**Control perturbation ablation (diffusion).** We perform an analogous shuffled-perturbation control in the diffusion setting. For each current level $I$, we reuse the targeted perturbation at the maximally face-selective location $y_{\text{face}} = 42$, randomly shuffle its entries across feature dimensions, and apply this shuffled perturbation to the unperturbed deepest-layer features. We then apply the same linear transformation to obtain a CLIP embedding and run Stable Diffusion with IP-Adapter using the same prompt $p$ and noise latent $\epsilon$. Comparing targeted and shuffled perturbations in this setting again controls for non-specific global modulation and tests whether the emergence of face-like structure depends on the spatially structured perturbation of face-selective regions.

## A.4 PASSIVE VIEWING DATA QUALITY

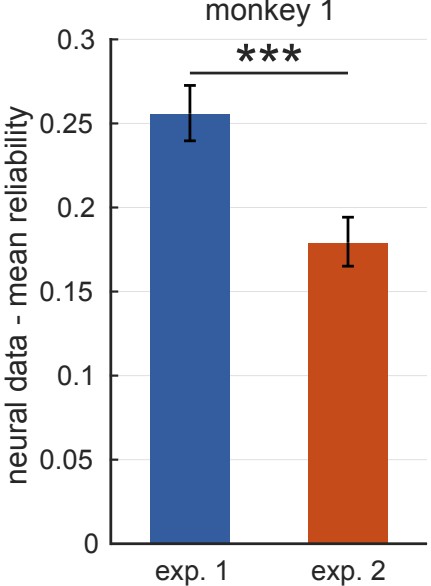

**Figure 10: Passive viewing data quality.** Using the passive viewing dataset from monkey 1, we quantified the reliability of each electrode following Papale et al. (2024). For each recording site, reliability was estimated by repeatedly splitting the 20 repetitions of each test image into two halves and computing the correlation between the resulting split-half response averages. Repeating this procedure yielded 190 reliability estimates per electrode, whose mean defines the electrode's reliability. At the population level (averaging reliability across electrodes), we observed a significant decrease from experiment 1 (February 2025) to experiment 2 (June 2025; $t = 6.84$, $p = 1.03 \times 10^{11}$, Cohen's $d = -0.30$), indicating a decline in overall signal stability. At the single-electrode level, 735 out of 1024 channels (72%) showed a significant reduction in reliability after Benjamini–Hochberg FDR correction. The mean within-electrode effect size (computed on the 190 split-half estimates per electrode) was large (Cohen's $d = -1.25$), reflecting a robust and widespread degradation of reliability over the four-month interval.

## A.5 USE OF LLMS

We used *ChatGPT (GPT-5, OpenAI, 2025)* to help polishing the writing of this document. The authors reviewed, edited, and are fully responsible for the final content.

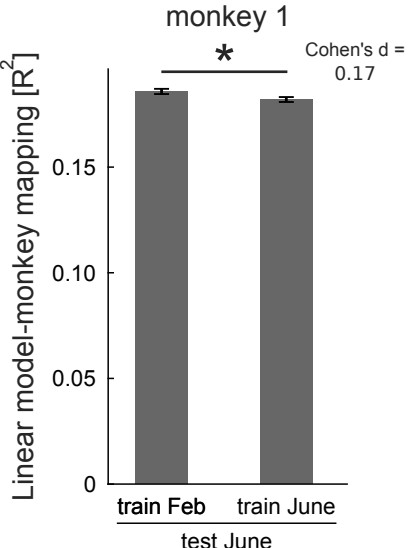

**Figure 11: Stability of model–brain alignment across months.** We quantified how stable the linear mapping between TDANN IT activations and neural responses remains over the four-month interval between February (experiment 1) and June (experiment 2) 2025 using the responses to the same passive viewing stimuli presented to monkey N. For each TDANN variant and seed, we fit a ridge-regression either on passive viewing data from February 2025 (experiment 1) or from June 2025 (experiment 2) and evaluated its predictions - in both cases - on the June data using 10-fold cross-validation with image splits shared across time points. The bars show the average variance explained ($R^2$) across all TDANN models, seeds, and folds when the mapping was trained on February data and tested on June data (left) versus when it was both trained and tested on June data (right). The February-trained mapping achieves $R^2 \approx 0.186$, compared to $R^2 \approx 0.182$ for the June-trained mapping, i.e. about $102\%$ of the variance explained by the within-session paradigm. Despite the decline in neural reliability documented in **Fig. 10**, the TDANN–IT mapping remains highly stable across months, suggesting that a single passive-viewing session can provide a model–brain alignment that remains usable over extended periods and thus reduces the need for repeated model-mapping sessions before each causal experiment.

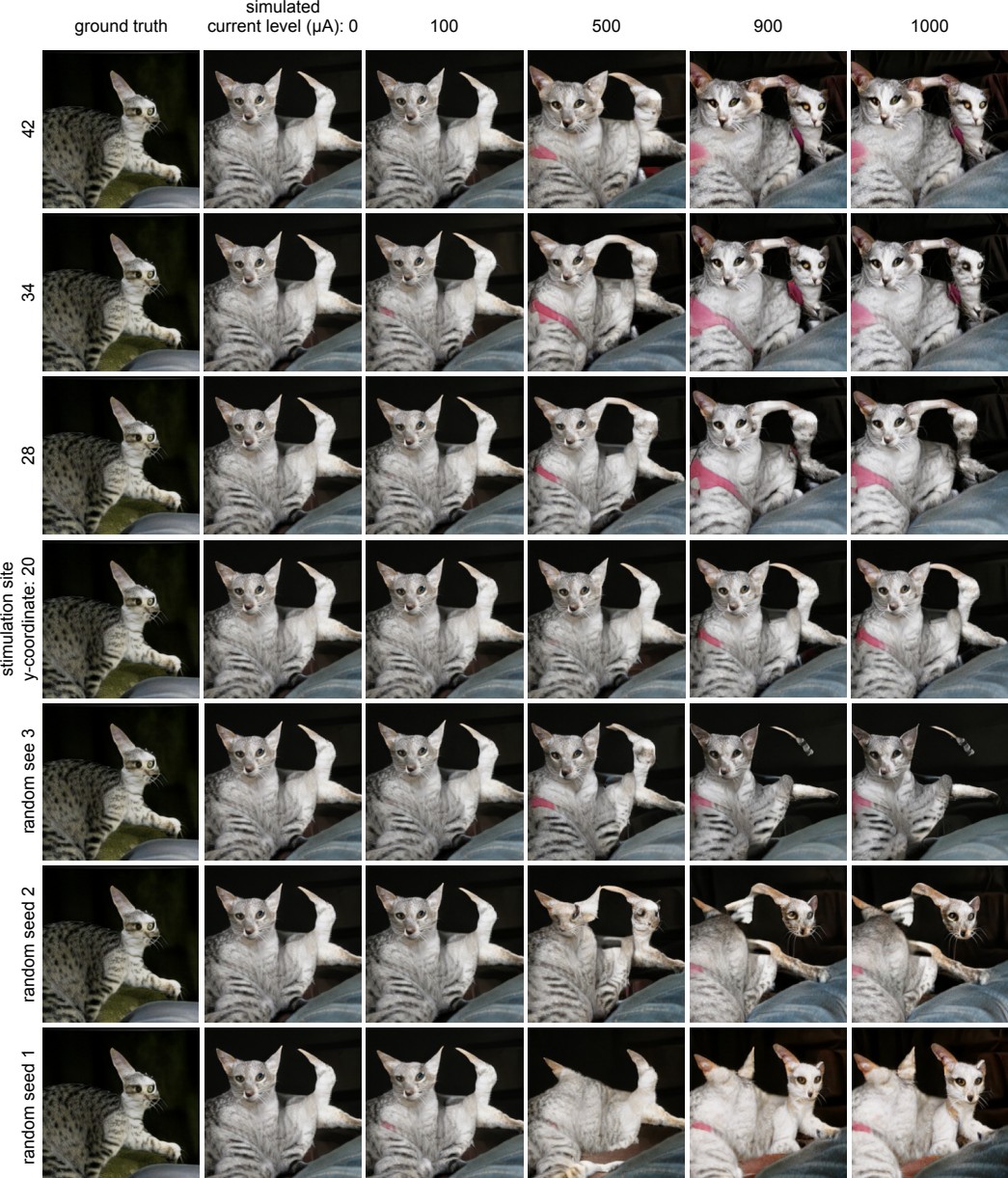

**Figure 12: Visualizing model perceptual effects of simulated microstimulation - image #1.**
Rows (from top to bottom): Y-coordinate in topographic model inferior temporal cortex (deepest layer) corresponding to high vs. low face-selectivity. Visualizing Columns: ground truth image randomly sampled from generative-adversarial-network (GAN) latent space, simulated current levels: $0, 100, 500, 1000\mu$A.

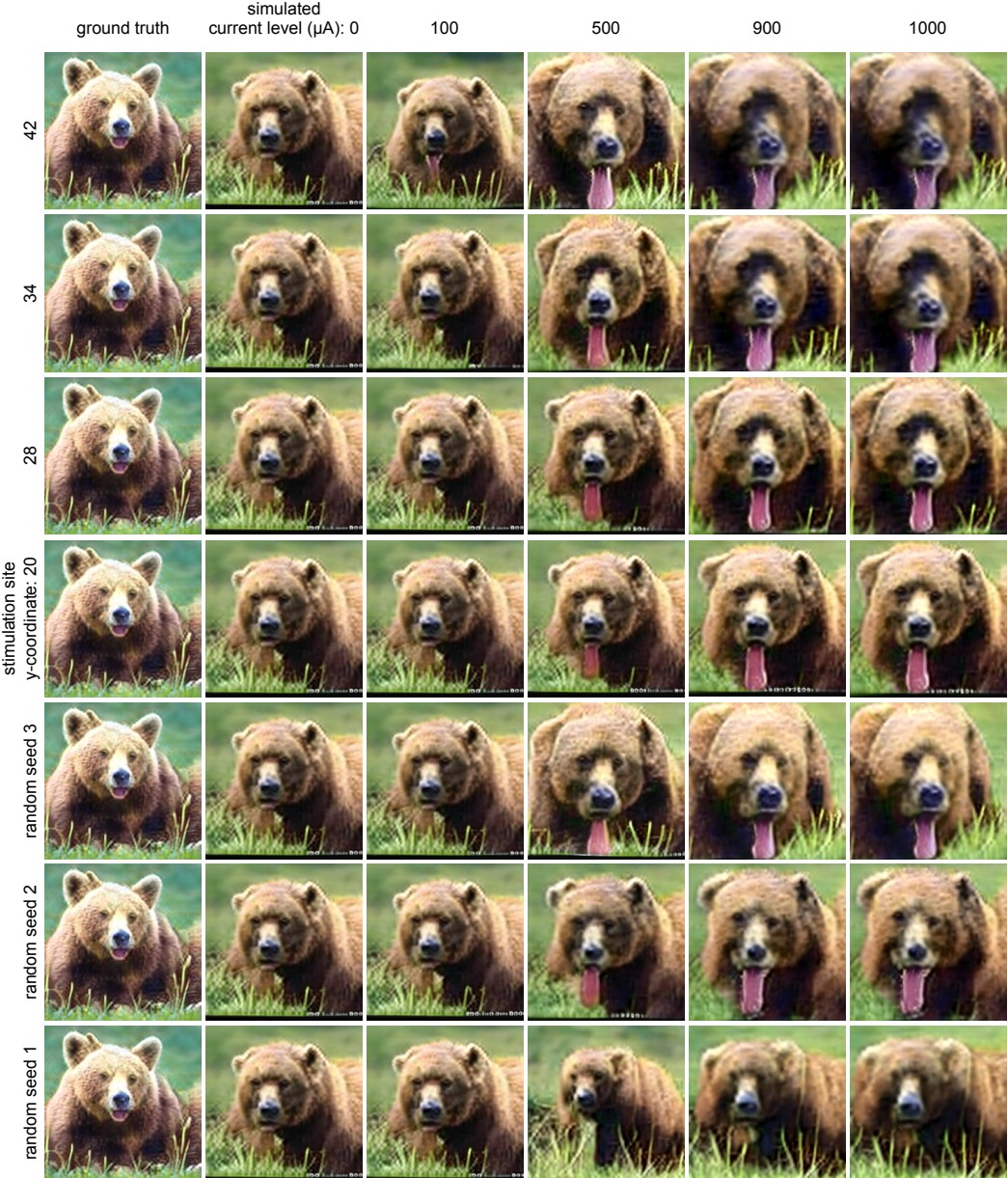

**Figure 13: Visualizing model perceptual effects of simulated microstimulation - image #533.**
Rows (from top to bottom): Y-coordinate in topographic model inferior temporal cortex (deepest layer) corresponding to high vs. low face-selectivity. Visualizing Columns: ground truth image randomly sampled from generative-adversarial-network (GAN) latent space, simulated current levels: $0, 100, 500, 1000\mu$A.

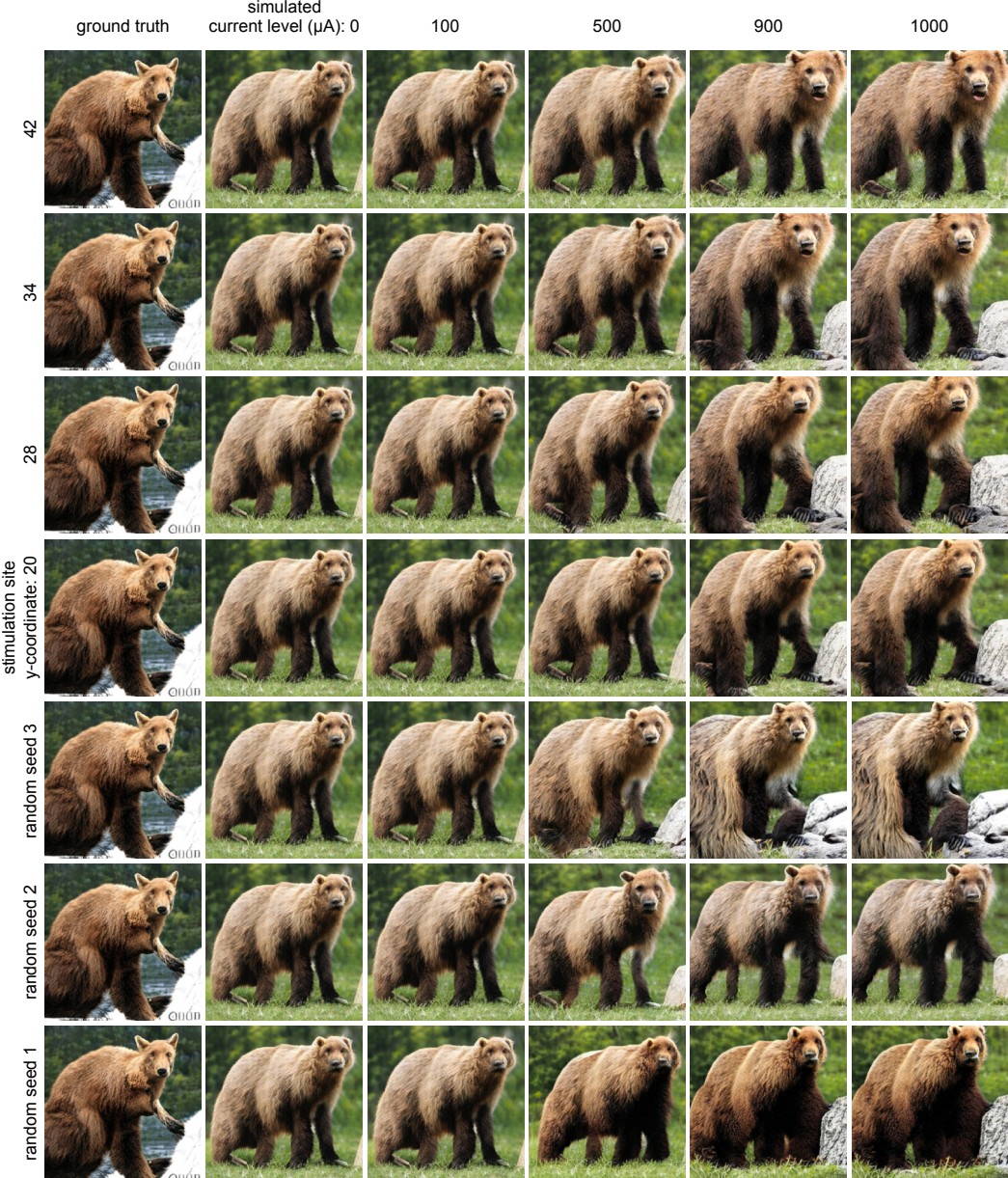

**Figure 14: Visualizing model perceptual effects of simulated microstimulation - image #530.**
Rows (from top to bottom): Y-coordinate in topographic model inferior temporal cortex (deepest layer) corresponding to high vs. low face-selectivity. Visualizing Columns: ground truth image randomly sampled from generative-adversarial-network (GAN) latent space, simulated current levels: $0, 100, 500, 1000 \mu$A.

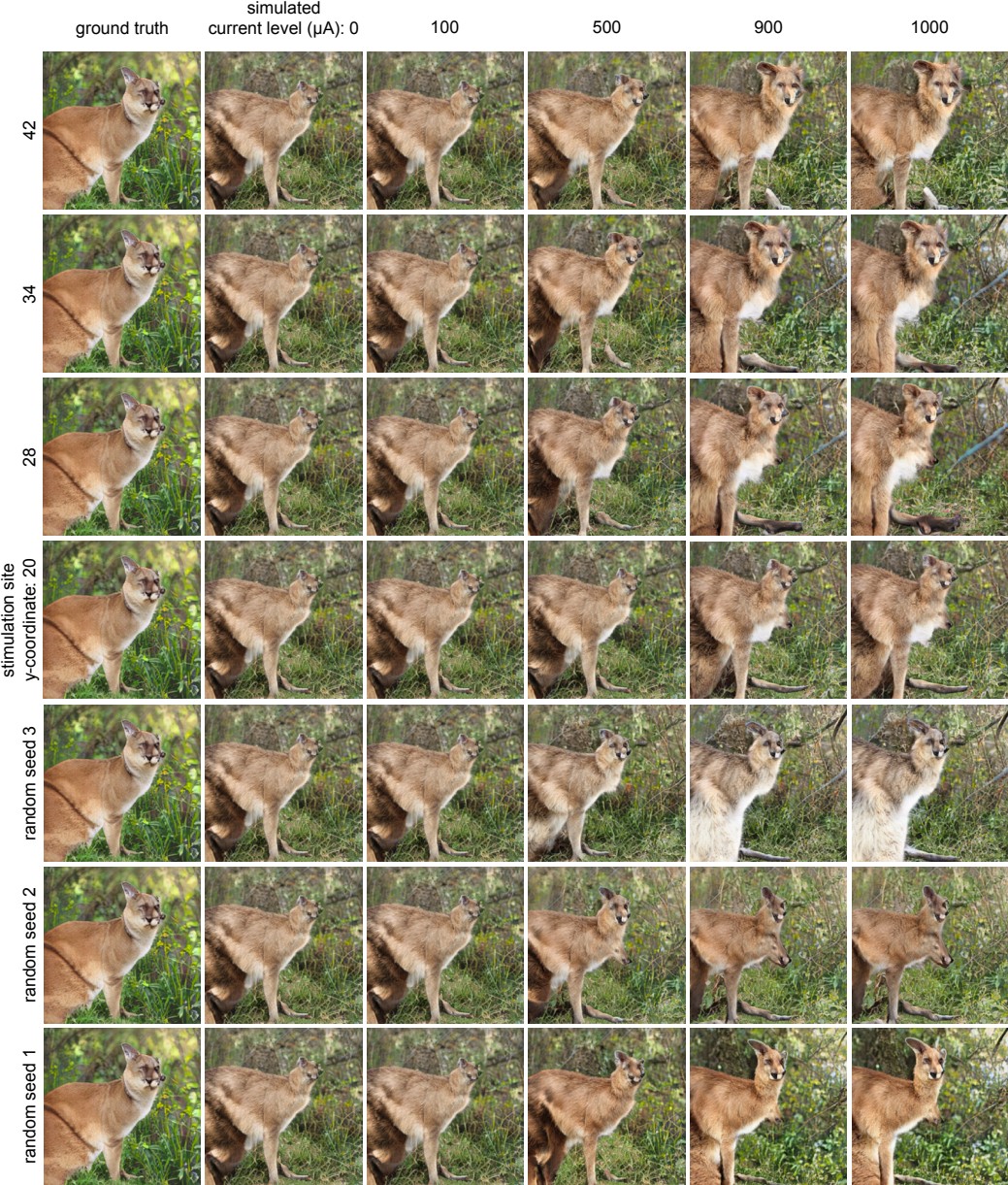

**Figure 15: Visualizing model perceptual effects of simulated microstimulation - image #3680.**
Rows (from top to bottom): Y-coordinate in topographic model inferior temporal cortex (deepest layer) corresponding to high vs. low face-selectivity. Visualizing Columns: ground truth image randomly sampled from generative-adversarial-network (GAN) latent space, simulated current levels: $0, 100, 500, 1000\mu$A.

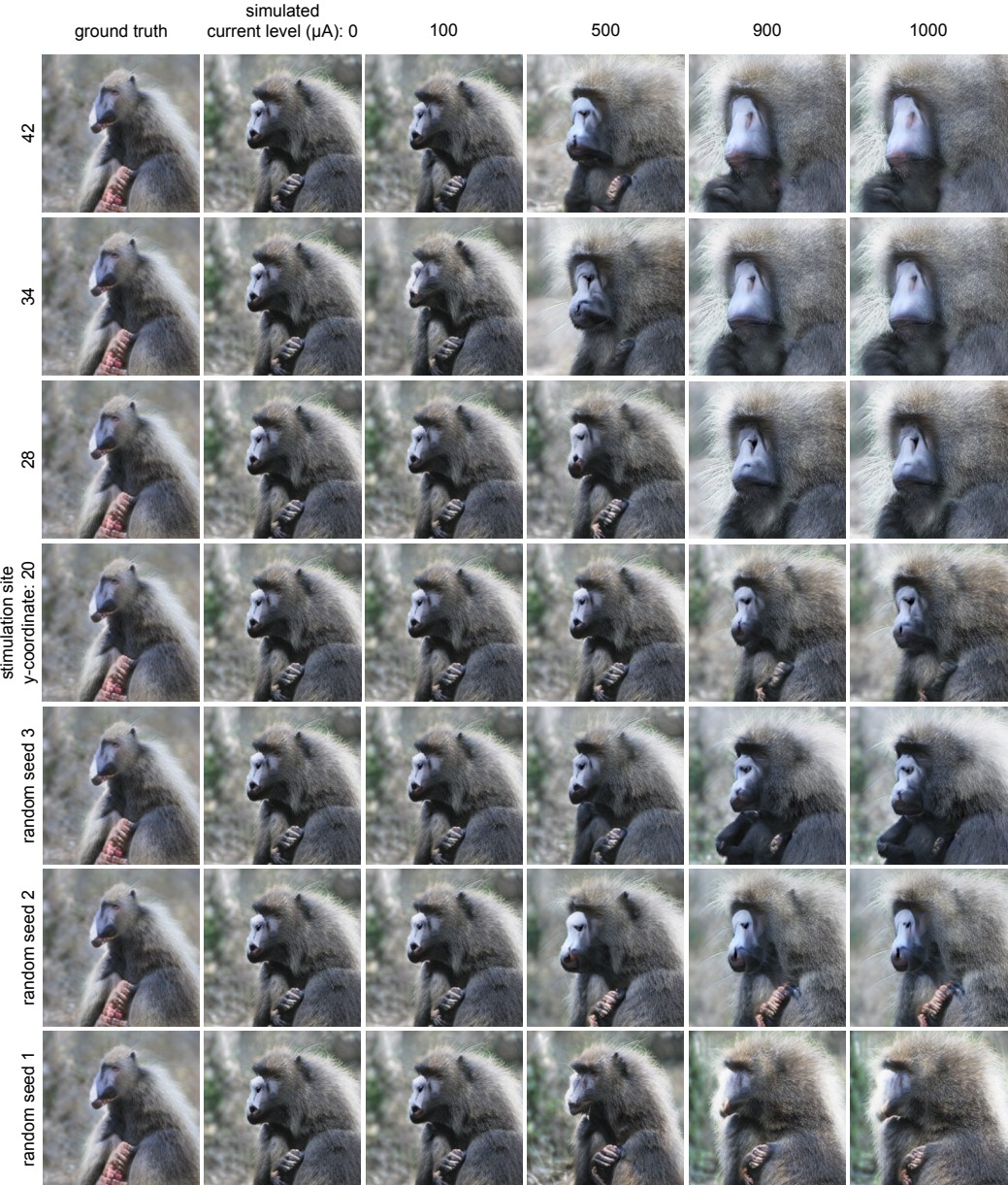

**Figure 16: Visualizing model perceptual effects of simulated microstimulation - image #1161.**
Rows (from top to bottom): Y-coordinate in topographic model inferior temporal cortex (deepest layer) corresponding to high vs. low face-selectivity. Visualizing Columns: ground truth image randomly sampled from generative-adversarial-network (GAN) latent space, simulated current levels: $0, 100, 500, 1000\mu$A.

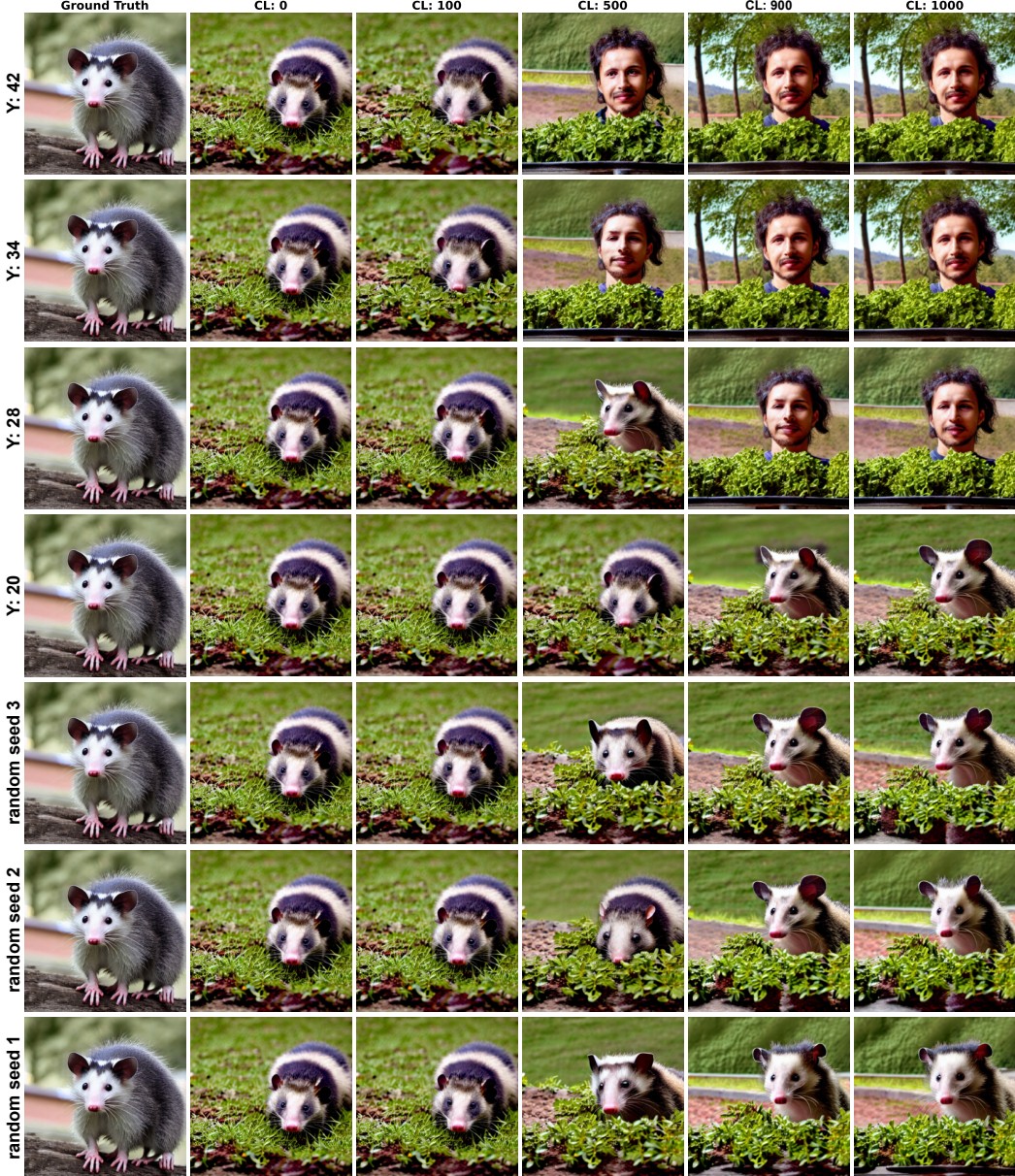

**Figure 17: Visualizing model perceptual effects of simulated microstimulation - image #1894.**
Rows (from top to bottom): Y-coordinate in topographic model inferior temporal cortex (deepest layer) corresponding to high vs. low face-selectivity (top to bottom).The last three rows are random stimulations of model IT that follow the distribution of multiplicative factors of a standard stimulation whose results are shown in the first 4 rows. Columns: ground truth image randomly sampled from diffusion model latent space, simulated current levels: $0, 100, 500, 1000\mu$A.

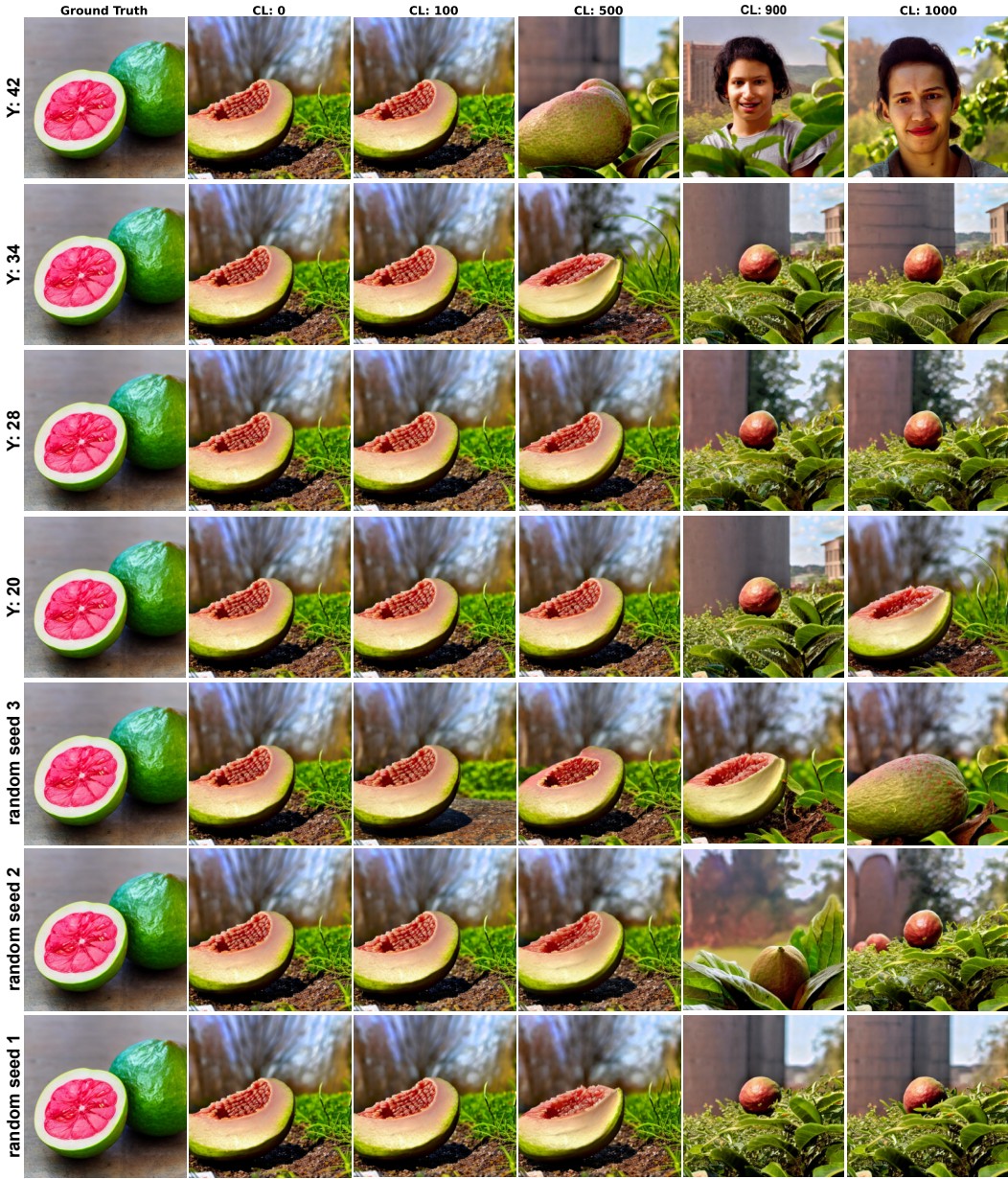

**Figure 18: Visualizing model perceptual effects of simulated microstimulation - image #2339.** Rows (from top to bottom): Y-coordinate in topographic model inferior temporal cortex (deepest layer) corresponding to high vs. low face-selectivity (top to bottom).The last three rows are random stimulations of model IT that follow the distribution of multiplicative factors of a standard stimulation whose results are shown in the first 4 rows. Columns: ground truth image randomly sampled from diffusion model latent space, simulated current levels: $0, 100, 500, 1000\mu$A.

