# OpenReview forum: "Model-Guided Microstimulation Steers Primate Visual Behavior"
_ICLR.cc/2026/Conference — ICLR 2026 Poster_

### Official Review · Reviewer_pN2f · 2025-10-20

**Soundness:** 3
**Presentation:** 4
**Contribution:** 4
**Rating:** 8
**Confidence:** 4

**Summary:**

The authors introduce a framework for guided microstimulation of the high-level visual cortex.  The authors use brain-aligned, topographic image-models as in-silico models of the brain, and prototype in-vivo microstimulation experiments based on in-silico observations.  Applying these prototype experiments in two macaque monkeys, the authors observe positive correlations between model-predicted behavior and observed monkey behavior.  The authors further report qualitative similarities between perceptions predicted by the model following microstimulation and those that have been previously reported by humans.

**Strengths:**

- Authors address an important open question in the neural engineering space: how to simulate neural stimulation of the visual cortex and perceptual outcome.
- All in-silico modeling is grounded in realistic (at least, most realistic for current state of the art) modeling of microstimulation (current spread, etc.) and cortical organization (through TDANNs)
- In experiment 1, positive correlations were observed between in-silico and in-vivo behavior changes with and without stimulations.  This provides evidence for the utility of such a closed-loop framework for studying perceptual effects of micro-stimulation.
- In experiment 2, stimulation parameters that were predicted to be effective by the modeling framework contributed to greater changes in AUC.  If results like this can be replicated in more studies, current work in microstimulation for visual perception could benefit greatly from this experimentation strategy.

**Weaknesses:**

- Simulating electrode placement and alignment with each model is unclear to me.  How are different positions of an array mapped onto a 512x7x7 latent space of a model?  Since each Utah array has 96/64 electrodes, does one spatial dimension of the latent space map to multiple electrodes?  What are the limitations of this?
- Correlation between model and monkey behavior diminishes in experiment 2.  Further discussion about potential reasons for this (e.g., is it because the TDANN is failing to predict activity accurately, too much noise in the in-vivo neural signal, etc.)  would be valuable.
- Qualitative visualizations of in-silico perceptual effects of perturbation are an interesting proof of concept but leave many open questions in terms of how perception should change with stimulation.  For example, in Figure 5, why does a second cat face appear in the picture of a cat while the single bear face enlarges in the bear picture (as opposed to the cat’s face enlarging or another face forming on the bear, for instance)?

**Questions:**

- Is it necessary that multiple models are used in the mapping procedure?  What is the variability in site predictivity between these different models?
- What was the average predictivity (e.g., correlation) between model-simulated neural site activity and measured in-vivo activity to the passive-viewing images? How do you think results may change when using models that are less/more predictive of neural activity in the brain?  Can this be quantified post-hoc?
- As suggested, TDANNs are seemingly critical for microstimulation modeling due to current spread to neighboring neurons.  How big of a role does this play in the models evaluated in this work?  Do non-TDANN models similarly have positive behavioral correlations with the monkeys (or with the TDANN models)? This supplemental analysis could quantitatively ground the choice for TDANN models in such a framework.

---

> ### Author Response · Authors · 2025-11-28
> **Response to questions 1**
>
> Thank you for your evaluation of our paper on using model-guided microstimulation to steer monkey behavior. You are raising important points that we respond to separately as follows.
>
> ### Questions
>
> **Predictivity of non-stimulation data. Is it necessary that multiple models are used in the mapping procedure? What is the variability in site predictivity between these different models?**
>
> Thank you for this important question. Linear predictivity across models, quantified as explained variance ($R^2$), ranged from [−0.06,0.27] for monkey 1 and [0.05,0.19] for monkey 2. Values close to zero likely reflect non-functional (“dead”) arrays with unreliable signal. Overall, this broad range indicates that model choice is indeed critical and that explicit model selection was necessary for successful model-guided stimulation. We have added a statement reporting these ranges and a more detailed description of the underlying cross-validated ridge regression in the last paragraph of Section 3.1.
>
> A definitive experimental answer would require testing a model with low predictivity on the corresponding array. However, since the stimulation devices have now been explanted due to reduced signal quality (Fig. 9), such a test is unfortunately not feasible at present.
> Predictivity of non-stimulation data. What was the average predictivity (e.g., correlation) between model-simulated neural site activity and measured in-vivo activity to the passive-viewing images? How do you think results may change when using models that are less/more predictive of neural activity in the brain? Can this be quantified post-hoc?
>
> For details on the effect of linear predictivity of a model based on the passive viewing data and its ability to predict the behavioral effects of microstimulation, please see our response to the previous question. Please let us know should this not answer your question.
>
>
> **As suggested, TDANNs are seemingly critical for microstimulation modeling due to current spread to neighboring neurons. How big of a role does this play in the models evaluated in this work? Do non-TDANN models similarly have positive behavioral correlations with the monkeys (or with the TDANN models)? This supplemental analysis could quantitatively ground the choice for TDANN models in such a framework.**
>
> Thank you for this suggestion. We will perform the same linear regression analysis using non-topographic control models and will report the range of linear predictivity values to relate them to the linear predictivity values observed with our topographic models.
>
>
> ### Other points raised by the reviewer
>
> **More detailed methods. Simulating electrode placement and alignment with each model is unclear to me. How are different positions of an array mapped onto a 512x7x7 latent space of a model? Since each Utah array has 96/64 electrodes, does one spatial dimension of the latent space map to multiple electrodes? What are the limitations of this?**
>
> Thank you for pointing out that our description of the in-silico cortex implemented as a single layer of a topographic network was not sufficiently clear. At the start of TDANN training, each unit in the layer is assigned a unique location on a 2D sheet, such that no two units occupy the same position. Thus, the 512 × 7 × 7 = 25,088 units are distributed across 25,088 distinct locations, forming an in-silico cortical sheet. Importantly, this spatial layout does not follow a Cartesian grid; instead, following Margalit et al. (2024), a spatial jittering procedure places all 512 units from a given location of the 7 × 7 grid on a distinct position on the sheet.
>
> This spatial organization serves two purposes: (1) during training, pairwise distances between units are used to compute the spatial loss that promotes smoothly varying functional maps (see Margalit et al., 2024; Rathi & Mehrer et al., 2025 for a language-model implementation), and (2) it enables distance-based computation between units and stimulation sites in the perturbation module that models the effects of microstimulation. We have clarified this in the first paragraph of Section 3.1.

---

> ### Author Response · Authors · 2025-11-28
> **Response to questions 2**
>
> **More detailed results section. Correlation between model and monkey behavior diminishes in experiment 2. Further discussion about potential reasons for this (e.g., is it because the TDANN is failing to predict activity accurately, too much noise in the in-vivo neural signal, etc.) would be valuable.**
>
> We believe that the decrease in correlation is mainly due to degrading signal. We computed the split-half reliability to estimate neural signal quality from the passive viewing data recorded before each experiment. The signal clearly deteriorates across the 4-month interval between experiment 1 and 2. We have included our computations in fig. 9 in the Appendix and included a statement on this issue in the first paragraph of section .
>
>
> **More detailed methods section - additional visualization methods. Qualitative visualizations of in-silico perceptual effects of perturbation are an interesting proof of concept but leave many open questions in terms of how perception should change with stimulation. For example, in Figure 5, why does a second cat face appear in the picture of a cat while the single bear face enlarges in the bear picture (as opposed to the cat’s face enlarging or another face forming on the bear, for instance)?**
>
> Each visualization method—GAN- or diffusion-based—has its own idiosyncrasies related to training data, the mapping from TDANN to the generative model, and other factors (see Figs. 5 and 10–14; Section 3.4 and Appendix A3). We therefore do not claim that these images represent “what the model sees,” but rather present them as visual intuition for TDANN representations.
>
> Guided by neuroscience findings, we probed stimulation sites with decreasing face-selectivity across different simulated current levels. This revealed consistent boosts in face-related image features with increasing face-selectivity and current level in both GAN- and diffusion-based visualizations (Figs. 10–16). However, this remains far from the systematic investigation required to fully address your question. For a more complete understanding, we would need to test additional stimulation sites and extend the analysis to other category-selective regions (e.g., scene-, tool-, or body-part selectivity). While we see strong merit in this direction and plan to pursue it for the camera-ready version, it goes beyond what is feasible for us within the rebuttal time frame.
>
> **In light of these substantial improvements of the manuscript, we would be grateful to the reviewer if she/he could reassess her/his evaluation of our paper.**

---

> ### Author Response · Authors · 2025-12-03
> **Response to questions 3**
>
> **Update on response to the reviewer’s question: “As suggested, TDANNs are seemingly critical for microstimulation modeling due to current spread to neighboring neurons. How big of a role does this play in the models evaluated in this work? Do non-TDANN models similarly have positive behavioral correlations with the monkeys (or with the TDANN models)? This supplemental analysis could quantitatively ground the choice for TDANN models in such a framework.”**
>
> Thank you for this suggestion. To directly address it, we trained a single non-topographic control model with the same ResNet-18 architecture, self-supervised objective, dataset (ImageNet-2012 + LFW), batch size, and 200-epoch training schedule as our TDANNs, but with the spatial loss switched off (α = 0). We then repeated the same linear-regression neural-predictivity analysis performed for the topographic models to assess the model-brain alignment. For monkey 1, the topographic models achieved $R^2$ values in the range 0.244–0.294 (mean 0.268) when using passive viewing data from February (experiment 1) and 0.135–0.241 (mean 0.182) when using data from June (experiment 2). The non-topographic control reached $R^2 = 0.27$ for the February session and $R^2 = 0.19$ for the June session on the same IT array. Thus, a model trained without spatial loss but under otherwise matched conditions achieves neural alignment similar to the topographic model we used in terms of linear predictivity.
>
> In light of this result and to answer your original question, we believe that the introduction of topography to the models does not strongly impact their brain alignment. It appears that there is no distinguishable difference with regard to model-brain alignment between our candidate topographic models and the one instance of a non-topographic model we were able to train for this control. We now report this additional analysis in the last but one paragraph in section 3.1.

---

### Official Review · Reviewer_cYUD · 2025-10-30

**Soundness:** 3
**Presentation:** 2
**Contribution:** 3
**Rating:** 6
**Confidence:** 5

**Summary:**

### Summary

The authors link the neuronal responses in monkey visual cortices to topographical ANN, and GAN latent space, and then using the in silico model to screen for effective experimental design that stimulation will have strong perceptual effect. Although the net perceptual effect of model predicted perturbations is not super strong, the model could in some case predict which image x perturbation site has stronger behavioral bias.

Although many components of the pipelines (Topographic ANN, perturbation model, GAN visualization, linking) have been established before, this is still a nice combination of all of them, and a stress test of all the components. Overall, it is a worthy addition to the literature, esp. the behavior modulation effect, and future works can build on top of it!

**Strengths:**

### Strength

- Predicting the perceptual effect of visual cortical stimulation is an important and hard problem in neuroscience, and one with significant translational application value.
- Many previous model optimized stimuli study only touches on manipulation of neuron firing, but this work advanced this direction in predicting the stimulation effects on behavior.
- The authors did a very comprehensive work with statistical rigor, and being very honest about many weaker effects and none effects. Kudos on that!

**Weaknesses:**

### Weakness

- **Clarity of the method**
I feel part of methods is too vague from the current main text.
    - Figure 6,7,8 are very clear and informative, moving some details from them to Figure 1 and Figure 2 will be great, currently Figure 1,2 are a bit too high level, and could be compressed a bit.
    - The topological ANN part is quite detailed in the text, maybe too detailed, but GAN, the regression method and other parts are much less so.
    - I feel mentioning and describing the task of monkey is quite important in the main text. Otherwise i don’t understand the stimulation, the sequence and perception paradigm.
- The behavior bias seems to be relatively small and non-robust. (Fig. 4) Although, we all know monkey behavior is very noisy and hard to get large effect size, so it is what it is.

**Questions:**

### Questions

- Why perform linking between CNN model and brain based on GAN images instead of natural image samples? is it because the available latent codes make the visualization easier?
- Which GAN did you use? There seems to be no details about it in the appendix. The architecture, training data etc. of GAN are critical and worth mentioning, since those are the prior of your visualization. Are they some StyleGAN2?
    - One reason I ask is that the latent space of many GANs have intriguing geometric structures [^0,^1], so in some cases perturbation along random ish direction could produce perceptually relevant change in images, as long as it has non zero inner product with those interpretable direction.
    - Because of it, you may want to do some random perturbation as control for Figure 5 visualization and see if they are still this interpretable.

[^0]: Voynov, & Babenko, (2020). Unsupervised discovery of interpretable directions in the gan latent space, ICML

[^1]: Wang, & Ponce, (2021). The geometry of deep generative image models and its applications. ICLR

- Regarding the link between GAN latents and visual neural code, i feel the authors could discuss more on the related works from the Ponce et al. threads. For example, the alignment of neuronal tuning in higher visual cortices with the latent space of different GANs (DeePSim & BigGAN), which allow the neurons to steer the latent codes of GAN and generate images that maximize their activations [^2, ^3]. I feel this is one of the bases for the GAN visualization method besides Papale 2024. In those approaches, they also found some behavioral effect of the images that optimize neuron activation in GAN space [^4].

[^2] Wang, Ponce, 2024, **Neural Dynamics of Object Manifold Alignment in the Ventral Stream** https://www.biorxiv.org/content/10.1101/2024.06.20.596072v1.abstract

[^3] Wang, B., & Ponce, C. R. (2022). Tuning landscapes of the ventral stream. *Cell reports* [https://www.cell.com/cell-reports/fulltext/S2211-1247(22)01460-7](https://www.cell.com/cell-reports/fulltext/S2211-1247(22)01460-7)

[^4] Rose, O., Johnson, J., Wang, B., & Ponce, C. R. (2021). Visual prototypes in the ventral stream are attuned to complexity and gaze behavior. *Nature communications*

- Why generating sequence of 7 images and requiring the image sequence to have monotonic effect? What’s the conceptual interpretation of it? Is that the authors are hypothesizing the effect of stimulation at various strength is equivalent to this sequence of image?
- What do we make of the null result in Sec. 4.2? Isn’t it that more stimulation sites should make a stronger correlation?
- Sec. 4.3 is very cool visualization, although we don’t have evidence that those are what monkeys actually perceive right? This is not an objection to the paper.
I think Afraz lab, has more extensive work testing the perceptual effect of stimulation, and it’s quite an expensive loop to validate these prediction (c.f. Perceptogram https://www.nature.com/articles/s41467-024-47356-8).
- Simulation of biological variability “To approximate biological variability and thus mimic trials, we therefore consider multiple GAN-generated sequences that are all optimized to modulate activity at the same monkey stimulation site. Each distinct sequence constitutes an in-silico trial, and averaging across these sequences provides a model analogue of the across-trial variability observed in the animal.”
    - It’s curious the authors simulate the neuron and choice variability by different image sequences generated by GAN. Admittedly this is a tractable way to generate variance, what’s the interpretation of it? Why not have some drop out / stochasticity in the topographical ANN?

**Details Of Ethics Concerns:**

N.A.

---

> ### Author Response · Authors · 2025-11-28
> **Response to questions 1**
>
> Thank you for your evaluation of our paper on using model-guided microstimulation to steer monkey behavior. You are raising important points that we respond to separately as follows.
>
> ### Questions
>
> **Why perform linking between CNN model and brain based on GAN images instead of natural image samples? Is it because the available latent codes make the visualization easier?**
>
> The established linear mapping is used to transform model activations associated with large predicted behavioral effects into neural activations expected to yield strong monkey behavioral responses. In our experiments, we used GAN-generated image sequences, as these were optimized to mimic the neural effects of microstimulation at targeted sites (see “Stimulus generation” in section 3.2). To remain within the same stimulus distribution, we also used images drawn from this GAN distribution to establish the model–monkey mapping.
>
> In principle, the mapping could have been based on naturalistic images, and the experimental stimuli could likewise have consisted of natural images. For example, one could select, for each stimulation site, a set of natural images that modulate the site’s response across a wide range. However, we have no clear indication of how this would have affected the accuracy of behavioral predictions. The primary motivation for using the GAN-based approach for both mapping and experimental stimuli was its demonstrated promise in combination with microstimulation in monkey IT (Papale et al. 2024).
>
>
> **Which GAN did you use? There seems to be no details about it in the appendix. The architecture, training data etc. of GAN are critical and worth mentioning, since those are the prior of your visualization. Are they some StyleGAN2?**
>
> Absolutely right, thank you for this suggestion. We have now added detailed descriptions of the GAN used to generate the 7-image stimulus sequences and the GAN-based visualization of microstimulation effects in model face-selective regions (new Section 3.4 and Appendix A3.1). In addition, we introduced an alternative visualization approach based on diffusion models and now describe this model in detail as well (Section 3.4 and Appendix A3.2).
>
> Thank you also for suggesting a control using random perturbations of the GAN latent space. We tested this by permuting the perturbation vector applied to topographic model IT (preserving its value distribution) and projecting the resulting activations into GAN and diffusion latent spaces (see Appendix A3.1 and A3.2). In the diffusion model, standard (unpermuted) perturbations produced stronger face-related changes with increasing current and face-selectivity (e.g., Fig. 15), whereas permuted perturbations induced comparable effects only at much higher current levels. Similarly, in the GAN-based visualizations, unpermuted perturbations at highly face-selective sites yielded stronger face-related effects than their permuted counterparts.
>
> While these results provide intuition about model IT representations, a more systematic analysis—extending beyond face-selectivity to other categories—will be required for stronger conclusions. We now explicitly note the exploratory and intuition-guiding nature of these visualizations at the end of Section 2.
>
> **Regarding the link between GAN latents and visual neural code, i feel the authors could discuss more on the related works from the Ponce et al. threads. For example, the alignment of neuronal tuning in higher visual cortices with the latent space of different GANs (DeePSim & BigGAN), which allow the neurons to steer the latent codes of GAN and generate images that maximize their activations. I feel this is one of the bases for the GAN visualization method besides Papale 2024. In those approaches, they also found some behavioral effect of the images that optimize neuron activation in GAN space.**
>
> Thank you for pointing out existing literature on using generative models to visualize representations in high-level visual cortex. We have now added a paragraph at the end of section 2 that includes Ponce 2019 and other related work.
>
>
> **Why generating sequence of 7 images and requiring the image sequence to have monotonic effect? What’s the conceptual interpretation of it? Is that the authors are hypothesizing the effect of stimulation at various strength is equivalent to this sequence of image?**
>
> Yes, exactly, the image sequence is optimized to mimic the effect of microstimulation at a targeted cortical site. Thank you for the hint, we have now made this connection clearer in section A.1 under “Ranking image sequences by a selectivity index. ”.

---

> ### Author Response · Authors · 2025-11-28
> **Response to questions 2**
>
> ### Questions
>
> **What do we make of the null result in Sec. 4.2? Isn’t it that more stimulation sites should make a stronger correlation?**
>
> From experiment 1 to 2, we reduced the spatial distance constraint between candidate stimulation sites to increase the likelihood of selecting sites predicted to yield strong behavioral effects (see updated Section 3.3). The primary goal was not to increase the absolute number of tested sites, but to raise the proportion of sites for which the model predicted strong behavioral changes. In experiment 1 we tested 13 sites in monkey 1 and 19 in monkey 2, whereas in experiment 2 we tested 9 sites in monkey 1 as now indicated in .
>
> Although the looser spatial constraint in experiment 2 could in principle have increased the model–behavior correlation, the absolute number of tested sites was in fact lower, likely due to declining signal quality (Appendix Fig. 9). We have clarified this distinction in the second paragraph of Section 4. Thank you for pointing this out.
>
>
> **Sec. 4.3 is very cool visualization, although we don’t have evidence that those are what monkeys actually perceive right? This is not an objection to the paper. I think Afraz lab, has more extensive work testing the perceptual effect of stimulation, and it’s quite an expensive loop to validate these prediction (c.f. Perceptogram https://www.nature.com/articles/s41467-024-47356-8).**
>
> While our visualizations can provide intuition about the representational changes induced by microstimulation in monkey IT, there is no principled way to determine whether any given image reflects “what the monkey saw” on a specific trial. The perceptogram approach developed by Arash Afraz’ lab comes closest to this goal. In this paradigm, subjects report whether stimulation occurred, and on a small subset of trials the stimulation is replaced by a visual stimulus optimized to elicit similar neural effects, leading the subject to believe they were stimulated. These trials therefore provide a behavioral proxy for the stimulation-induced percept.
>
> As you noted, this approach is experimentally demanding, as such test trials must remain rare to avoid expectation effects, requiring very large numbers of overall trials. We have now included perceptograms into our list of related work at the end of Section 2.
>
>
> **Simulation of biological variability “To approximate biological variability and thus mimic trials, we therefore consider multiple GAN-generated sequences that are all optimized to modulate activity at the same monkey stimulation site. Each distinct sequence constitutes an in-silico trial, and averaging across these sequences provides a model analogue of the across-trial variability observed in the animal.”. It’s curious the authors simulate the neuron and choice variability by different image sequences generated by GAN. Admittedly this is a tractable way to generate variance, what’s the interpretation of it? Why not have some drop out / stochasticity in the topographical ANN?**
>
> Thank you for your question regarding the introduction of variability in our experimental setup. We could have introduced variability via dropout or by adding noise to activations, thereby creating a notion of trials in the model. Instead, we chose to use multiple GAN-generated image sequences (optimized to suppress or drive the same cortical site) to induce variability. The main motivation was to ensure that our data can easily serve as a benchmark: when variability is introduced through distinct image sequences, other topographic models can be tested without any modification. In contrast, introducing variability through dropout or noise would require model-specific adaptations with additional parameters governing noise type and strength. By placing variability on the data side, different models can therefore be compared more straightforwardly. We have added this clarification at the end of Section A.2.2.

---

> ### Author Response · Authors · 2025-11-28
> **Response to questions 3**
>
> ### Other points raised by the reviewer
>
> **More detailed methods section. Clarity of the method I feel part of methods is too vague from the current main text. Lots of details on topographic ANNS, but GAN and regression (mapping) method lack detail**
>
> Thank you for this suggestion, we have made the following changes to the methods section.
>
> - In the last paragraph of section 3.1 you can now find more details on the model-monkey mapping procedure based on linear predictivity including the range of explained variance across the combinations of models and monkey arrays.
> - While we appreciate your suggestion to put some of the content of the appendix figures 6,7, and 8 into the first two figures, we prefer keeping the higher-level overviews in figures 1 and 2 as it allows readers less familiar with the subject to orient themselves more easily before diving into more detailed, but important descriptions of e.g. the experimental design. Instead, we have moved some of the description of the monkey behavioral task to the main text (see updated section 3.3) and hope this facilitates the reading flow.
> - In section 3.4 and Appendix section A3.1, one can now find a more detailed description of the GAN.
>
> **In light of these substantial improvements of the manuscript, we would be grateful to the reviewer if she/he could reassess her/his evaluation of our paper.**

---

### Official Review · Reviewer_LktB · 2025-10-30

**Soundness:** 3
**Presentation:** 3
**Contribution:** 3
**Rating:** 6
**Confidence:** 4

**Summary:**

The paper proposes a practical pipeline for model-guided microstimulation of macaque inferior temporal (IT) cortex that links: 1) a spatially explicit perturbation module that translates microstimulation parameters to local activity changes, 2) topographic DNNs aligned to each animal’s IT array via passive-viewing data, and 3) a mapping that projects model-optimized sites back to electrodes for in-vivo testing. In Experiment 1, per-site model predictions of behavioral bias in a 2AFC recognition task correlate with stimulation-evoked shifts in both monkeys, although mean shift is not above zero. In Experiment 2, the authors observe a significant stimulation-driven bias in one monkey but the per-site correlation vanishes. They also visualize in-silico "facephene-like" effects by mapping perturbed model states into a GAN latent space. Together this is pitched as proof-of-principle that topographic models can steer causal interventions in higher-level vision.

**Strengths:**

* Originality: Introduces a prospective, model-in-the-loop pipeline that uses animal-specific topographic alignment to guide microstimulation in IT. Integrates a spatial perturbation module with a topographic DNN and adds generative reconstructions for interpretable inspection. This combination is new in the context of causal tests in higher visual cortex.
* Quality: Implements a clear alignment procedure on passive-viewing data, then prospectively tests model-chosen site-stimulus pairs in primates. Reports statistics transparently and separates exploratory modeling from in-vivo confirmation. The overall methodology is careful and reproducible in principle.
* Clarity: Explains the 3-stage workflow with readable figures and step-wise descriptions. The mapping from electrodes to model units and the procedure for stimulus selection are described in sufficient detail to follow.
* Significance: Provides a practical blueprint for using topographic models to steer causal interventions in the ventral stream. Establishes a path toward systematic, theory-driven stimulation studies.

**Weaknesses:**

* The paper does not deliver both a significant mean effect and reliable per-site predictivity within the same experimental setting. This weakens the claim that the same model instance robustly predicts and drives behavior
* The authors report r values but not R2. For the combined correlation r=0.53, the model explains about 28% of the variance in stimulation-evoked behavioral shifts. This should be stated with confidence intervals and compared to baselines. There is also no estimate of the fraction of neural variance captured by the perturbation module in IT units
* The current-distance form is sensible, but the paper does not report sensitivity analyses over decay constants, gain factors, clipping, or non-linearities that are known to matter in IT microstimulation. Readers need to know whether the observed behavior is robust to these choices
* Helpful for intuition, but there is no quantitative link to percept reports or neural selectivity beyond anecdotal examples
* Stimulation of IT is unlikely to yield stable, retinotopically anchored phosphenes or letter-like percepts. At best it may bias category representations, consistent with facephene overlays, which limits near-term translatability to clinical visual prostheses that aim for placeable, compositional percepts. The paper should temper scope and clarify that this is about biasing recognition, not restoring structured visual qualia

**Questions:**

* What fraction of observed neural variance at and around the stimulated sites is captured by the perturbation module when fitted to pre-stimulation passive-viewing data?
* How stable is the model-brain alignment across days. If the passive-viewing alignment is from 2 to 4 days prior, what is the drop in predictivity if reused without re-alignment?
* Can you include ablations over decay constant, gain, clipping rmax, additive vs multiplicative perturbations ...? Which parameters most strongly affect predicted \Delta AUC?
* For Experiment 2, do you have sham-only and random-site controls matched in number of trials? Was there a pre-registered stopping rule or power analysis?
* Please clarify the intended prosthetic use case. IT stimulation may bias recognition choices rather than evoke stable object percepts. Can you propose measurable milestones that connect this method to clinically meaningful endpoints?

---

> ### Author Response · Authors · 2025-11-28
> **Response to questions**
>
> Thank you for your evaluation of our paper on using model-guided microstimulation to steer monkey behavior. You are raising important points that we respond to separately as follows.
>
> ### Questions
>
> **Control conditions. Can you include ablations over decay constant, gain, clipping rmax, additive vs multiplicative perturbations ...? Which parameters most strongly affect predicted ΔAUC?**
>
> Thank you for suggesting an exploration of behavioral effects across the parameter space of the perturbation module. The parameters were chosen based on the limited set of in-vivo studies investigating neural effects of microstimulation (Stoney et al., 1968; Histed et al., 2009; Majaj et al., 2015; Kumaravelu et al., 2022), following the formulation used in Schrimpf et al. (2024; see “Perturbation modules,” section 3.2). Our primary goal was to employ a perturbation module that has been shown to work in practice—i.e., one that allowed models to qualitatively predict behavioral outcomes across multiple causal intervention studies (Schrimpf et al. 2024)—and test whether such a model-guided approach can steer monkey behavior in vivo.
>
> We agree that assessing robustness to alternative perturbation implementations is an important direction, but due to declining signal quality (see Fig. 9) the recording and stimulation arrays had to be explanted in both animals. Re-implantation will take several months, making the requested in-vivo tests infeasible within the rebuttal timeframe. We therefore can unfortunately not evaluate how varying perturbation parameters affects monkey behavior at this stage, but consider this an important target for future work.
>
>
> **Control conditions. How stable is the model-brain alignment across days. If the passive-viewing alignment is from 2 to 4 days prior, what is the drop in predictivity if reused without re-alignment?**
>
> The 2–4 day interval spans a time window that, in our experience, has proven reliable across subjects and experimental paradigms. To address this question more quantitatively, we will apply the linear mapping derived from passive viewing data in experiment 1 (February 2025) to predict passive viewing responses in experiment 2 and will report the resulting ranges of linear predictivity values ($R^2$) for within-experiment versus cross-experiment predictions.
>
> While this analysis will provide useful insights, the more definitive question is how well- versus poorly-aligned models affect monkey behavior. Unfortunately, this cannot currently be tested, as the explantation of recording and stimulation devices in both animals prevents additional in-vivo experiments, at least within the rebuttal timeframe.
>
>
> **Control conditions. For Experiment 2, do you have sham-only and random-site controls matched in number of trials? Was there a pre-registered stopping rule or power analysis?**
>
> Sham and stimulation trials were matched in number and presented in pseudo-random order to minimize expectations regarding stimulation versus sham. To more clearly demonstrate that the observed effects were specific to model-selected sites, we had planned additional controls such as stimulating randomly chosen sites. However, due to degrading signal quality (see Fig. 9), these follow-up investigations were unfortunately not feasible.
>
>
> **Predictivity of non-stimulation data. What fraction of observed neural variance at and around the stimulated sites is captured by the perturbation module when fitted to pre-stimulation passive-viewing data?**
>
> Thank you for this question. To enable model and array selection, we quantified linear predictivity ($R^2$) between the model’s deepest layer (model IT) and all electrodes of each Utah array. Across models, values ranged from [−0.06,0.27] for monkey 1 and [0.05,0.19] for monkey 2. Values close to zero likely reflect non-functional (“dead”) arrays with unreliable signal. This broad range indicates that model choice is indeed critical and that explicit model selection was necessary for successful model-guided stimulation in both animals. We have added a statement reporting these ranges and a more detailed description of the cross-validated ridge regression procedure in the final paragraph of Section 3.1.
>
> A definitive experimental answer would require selecting a model with low predictivity and testing it on the corresponding array. However, as the stimulation devices have now been explanted due to declining signal quality (Fig. 9), such a control experiment is unfortunately not feasible at the moment.

---

> ### Author Response · Authors · 2025-11-28
> **Response to questions 2**
>
> ### Questions
>
> **Scope. Please clarify the intended prosthetic use case. IT stimulation may bias recognition choices rather than evoke stable object percepts. Can you propose measurable milestones that connect this method to clinically meaningful endpoints?**
>
> Our main claims (Figs. 3 & 4) concern monkey behavior and do not directly address the underlying representational changes that give rise to these effects. We leave open whether the induced neural changes should be interpreted as biasing category representations rather than restoring “visual qualia.” However, we believe model-guided microstimulation is sufficiently versatile to plausibly enable more fine-grained manipulation of visual percepts.
>
> For example, given the existence of eye-selective patches within face-selective regions \citep{Issa2012}, it is conceivable that additional subregions selective to other facial features exist and that differential stimulation of such regions could modulate perception along continuous face-feature dimensions (e.g., eye size, nose shape, mouth configuration), rather than merely inducing categorical shifts. Even if clearly separable face-feature patches do not exist, IT face regions likely encode more abstract facial dimensions that could support detailed perceptual modulation beyond category-level changes.
>
> More broadly, if it were technically feasible to causally access all IT neurons, it does not seem implausible that perceptual changes beyond categorical shifts could be evoked. We therefore view it as an empirical question whether IT activations are not only necessary but also sufficient for detailed visual percepts, or whether stimulation of earlier visual areas is required. We have expanded the final paragraph of the Discussion to reflect this perspective.
>
> ### Other points raised by the reviewer
>
> **The authors report r values but not R2. For the combined correlation r=0.53, the model explains about 28% of the variance in stimulation-evoked behavioral shifts. This should be stated with confidence intervals and compared to baselines. There is also no estimate of the fraction of neural variance captured by the perturbation module in IT units.**
>
> Thank you for the suggestion to increase the level of detail in the description of our main results. Section 4.1 now includes confidence intervals and a shuffled baseline distribution for comparison.
>
> We were not fully sure what you meant by the fraction of neural variance in IT units explained by the perturbation module. Could you please clarify this point? Thank you.
>
> **Helpful for intuition, but there is no quantitative link to percept reports or neural selectivity beyond anecdotal examples.**
>
> The visualizations we provide (section 3.4 and appendix section 3) are all reconstructions of the topographic model IT representations. We do not claim that the images are “what the model sees” but reconstructions thereof, and provide the images as a visual intuition of the TDANN representations. Thank you for pointing out the anecdotal nature of the visualization. Our approach may provide intuitions about model IT representations, but a more systematic exploration also including selectivity maps to other categories than faces will be needed to draw clearer conclusions. We have included a statement about the intuition-related nature of our visualization approach at the end of the last paragraph in section 2
>
>
> **In light of these improvements of the manuscript, we would be grateful to the reviewer if she/he could reassess her/his evaluation of our paper.**

---

> ### Author Response · Authors · 2025-12-02
> **Response to questions 3**
>
> **Update to response on the reviewer's question: “Control conditions. How stable is the model-brain alignment across days. If the passive-viewing alignment is from 2 to 4 days prior, what is the drop in predictivity if reused without re-alignment?”**
>
> As announced in our previous response from Friday, 28 Nov 2025, we quantified the stability of the linear model-brain alignment across days using the StyleGAN-XL passive-viewing data from monkey N, where the same set of 4,000 images was presented in two sessions (February and June 2025). For each TDANN variant, we fit a ridge regression from TDANN IT activations to the monkey multi-unit activity using 10-fold cross-validation and compared the two conditions most relevant for the stimulation experiments: 1. training on the February data and testing on the June data (train-Feb→test-June), and 2. both training and testing on the June data with identical image splits (train-June→test-June).
>
> Across all TDANN models and seeds, the February-trained mapping achieved an average predictivity of $R^2=0.186$ when evaluated on the June session, compared to $R^2=0.181$ for the June-trained mapping—i.e., the earlier mapping captures about 102% of the variance explained by the within-session mapping on the same June data. This shows that the learned linear mapping between TDANN and IT remains highly stable even over a four-month interval.
>
> We appreciate the reviewer’s suggestion to quantify this stability, as the result directly informs our experimental practice: it indicates that a single passive-viewing session may be sufficient to obtain a model–monkey alignment that remains valid for months, reducing the need to reacquire alignment data before each causal experiment. Since the stimulation experiments in our study used only a 2–4 day interval between alignment and intervention, the stability demonstrated here is more than adequate. We have now included this analysis in the Appendix Fig. 10).

---

> ### Author Response · Authors · 2025-12-02
> **Response to questions 4**
>
> **Update on response to the reviewer's question: “The authors report r values but not R2. For the combined correlation r=0.53, the model explains about 28% of the variance in stimulation-evoked behavioral shifts. This should be stated with confidence intervals and compared to baselines. There is also no estimate of the fraction of neural variance captured by the perturbation module in IT units”**
>
> With regard to the “fraction of neural variance captured by the perturbation module in IT units,” we interpret this as asking how much variance in IT activity the model accounts for at the neural level. In our experiments, however, the perturbation module is not fit to neural data recorded during stimulation—its parameters come from prior work (Schrimpf et al., 2024) and from the microstimulation literature (Section 3.2). Moreover, we could not record neural activity during stimulation because our devices do not support simultaneous recording, and even if they did, stimulation artifacts make such measurements technically non-trivial. We therefore cannot report a neuron-level R² specifically for the perturbation module.
>
> What we can quantify and have quantified is how much variance model IT explains in real IT activity during passive viewing, where reliable estimates are possible. Across TDANN variants and IT arrays, cross-validated linear predictivity ranges from −0.06 to 0.27 for monkey 1 and from 0.05 to 0.19 for monkey 2 (updated Section 3.1). This quantifies the degree of model–brain alignment at the neural level.
>
> The perturbation module is then validated indirectly through behavior: the model explains ≈28% of the variance in stimulation-evoked behavioral shifts (r = 0.53), now reported with confidence intervals and a shuffled baseline in Section 4.1.

---

### Official Review · Reviewer_GiCt · 2025-11-01

**Soundness:** 2
**Presentation:** 2
**Contribution:** 3
**Rating:** 4
**Confidence:** 4

**Summary:**

This paper introduces a model-guided microstimulation framework that combines topographic deep neural networks (TDANNs), in-silico perturbation modeling, and primate behavior testing to steer visual recognition behavior in macaques. The authors target higher-level visual cortex (IT) instead of early visual areas to evoke complex object-level percepts, bypassing the limitations of traditional retinotopy-based prosthetics. They demonstrate that model-predicted stimulation sites and image sequences can bias perceptual choices in a 2AFC task, and that simulated stimulation of face-selective sites qualitatively reproduces human-like facephenes. This is a proof-of-concept for closed-loop, model-driven neural intervention in vision.

**Strengths:**

Originality: First to use topographic ANNs to prospectively guide microstimulation in IT cortex, moving beyond retinotopy-based V1 stimulation.

Quality: Rigorous model-brain alignment, GAN-based stimulus generation, and behavioral validation with correlation analysis and effect size reporting.

Clarity: Extremely well-structured, with intuitive visuals, clear task design, and accessible neuroscience context.

Significance: Opens a new path for next-generation visual prosthetics targeting object-level percepts, and establishes a model-in-the-loop framework for causal neural control.

**Weaknesses:**

1.Sample size & signal degradation: Only 2 monkeys, with one excluded from Exp 2 due to signal loss, limiting statistical power and generalizability.

2.Model correlation drops in Exp 2: While behavioral bias is induced, the per-site correlation between model and monkey behavior disappears (r = 0.09), raising questions about robustness and model generalization.

3.Lack of control ablation: No random stimulation baseline or non-selective site control to rule out generic attention or arousal effects.

4.Stimulation parameters fixed: Only one current level (50 μA) and single-site stimulation tested; no dose-response or multi-site exploration.

5.Model variability not explored: Only one TDANN instance is used for mapping; no ensemble or uncertainty quantification across model initializations.

**Questions:**

1.Why did the model-behavior correlation collapse in Exp 2? It seems that the experimental results are not satisfactory. What were the author's remedial measures?

2.Was any other stimulation baseline run? The choice of the baseline of "no stimulation" in this article is too weak. It is insufficient to prove the effectiveness of the method proposed in this article.

3.How do you rule out that the observed bias is not due to generic attention or arousal from microstimulation? Welcome the author to share their thoughts.

4.Why not test multiple current levels or multi-site stimulation?

5.How sensitive are results to model choice?

6.Did you test other TDANN architectures or ensemble predictions to assess robustness?

---

> ### Author Response · Authors · 2025-11-28
> **Response to questions 1**
>
> Thank you for your evaluation of our paper on using model-guided microstimulation to steer monkey behavior. You are raising important points that we respond to separately as follows.
>
> ### Questions
>
> **Control conditions. Was any other stimulation baseline run?**
>
> Thank you for this suggestion, which is very much in line with our plans for additional in-vivo experiments. You are pointing out that, to attribute the observed effects exclusively to our experimental manipulations, it would be important to include baseline tests that rule out alternative explanations. We fully agree, and had in fact planned the following control conditions once the main effect was established.
> First, we aimed to show that randomly selecting a stimulation site and pairing it with a randomly selected 7-image sequence would neither produce a correlation between model and monkey behavior (as observed in experiment 1), nor drive monkey behavioral responses significantly above zero (as observed in experiment 2). Second, in a more controlled test, we planned to select site–sequence combinations that the model explicitly predicts to have no effect and verify that these do not yield significant behavioral changes in the monkey.
>
> Due to degrading signal quality (see Fig. 9) and the subsequent explantation of the Utah arrays in both animals, we were unable to run these controls in vivo. They therefore remain key target experiments for future work.
>
>
> **Control conditions. Why not test multiple current levels or multi-site stimulation?**
>
> Similar to the preceding question, you are rightfully requesting adequate baselines. Testing multiple current levels was indeed one of the key parameters we had planned to explore. However, as with the other control conditions, degrading signal quality over time (for details, see Fig. 9) and the subsequent explantation of the recording and stimulation devices in both animals prevented us from performing these tests in vivo.
>
> Multi-site microstimulation has, to our knowledge, not yet been systematically explored in published monkey studies and, as we note in the Discussion, represents an important target for future work. At the same time, it goes beyond what we could realistically investigate within the constraints of the current dataset and experimental setup.
>
>
> **Control conditions. Did you test other TDANN architectures or ensemble predictions to assess robustness?**
>
> The only model class we explored was the TDANN architecture from Margalit et al. 2024 trained on different combinations of image sets. We will explore other topographic vision models in the future (e.g. Lu et al., 2023, Deb et al., 2025) and have now included this plan into our discussion section. Thank you for the hint.
>
>
> **Control conditions. How sensitive are results to model choice?**
>
> Thank you for this question. The linear predictivity across models, quantified as explained variance ($r^2$), ranged from –0.06 to 0.27 for monkey 1 and from 0.05 to 0.19 for monkey 2. Values near zero likely correspond to arrays whose signal was no longer usable. These broad ranges indicate that model choice matters and that our model-selection step is indeed necessary for successful model-guided stimulation. We have now added a statement on this variability—together with a more detailed description of the underlying cross-validated ridge regression—to the final paragraph of Section 3.1.
>
> To conclusively answer your question with experimental evidence, we would need to select a model with low linear predictivity for a given array and run the stimulation experiment using that pairing. Because the stimulation devices have since been explanted due to the signal degradation described in Fig. 9, such a test is unfortunately not currently feasible.

---

> ### Author Response · Authors · 2025-11-28
> **Response to questions 2**
>
> ### Questions
>
> **More detailed results section. Why did the model-behavior correlation collapse in Exp 2?**
>
> As the reviewer correctly notes and as discussed in Sections 4.1 and 4.2, the correlation between model-predicted and observed monkey behavior decreases from a robust value in experiment 1 (combined across monkeys, $r = 0.58$, $p < 0.01$) to a non-significant correlation in experiment 2 (remaining single subject, $r = 0.09$). After experiment 1, our goal was to increase effect size by loosening the spatial constraint on the distance between neighboring stimulation sites within the array, thereby increasing the number of candidate sites tested. This strategy was partially successful: whereas monkey behavioral responses were not significantly different from zero in experiment 1 (Wilcoxon signed-rank test, $p > 0.05$), they became significantly biased in experiment 2 (Wilcoxon signed-rank test, $p = 0.043$).
>
> At the same time, as the reviewer correctly highlights, this improvement in overall behavioral effect coincided with a loss of correlation between model and monkey behavior. We believe this is primarily due to degrading signal quality between experiments rather than a failure of the model-guided optimization procedure. Indeed, passive viewing data recorded before both experiments in monkey 1 show a marked decrease in split-half reliability over the four-month interval (see Appendix Fig. 9). We have updated the second paragraph of Section 4 to clarify the motivation and consequences of these experimental changes. Thank you for raising this important point.
>
>
> **How do you rule out that the observed bias is not due to generic attention or arousal from microstimulation?**
>
> Our experimental design minimizes expectation effects by randomly interleaving stimulation and non-stimulation trials (50% / 50%), making it unlikely that systematic expectations drive the observed behavior via attention or arousal. We cannot fully exclude the possibility that stimulation itself induces arousal or attentional changes, as we did not directly measure these factors. However, while such effects are well documented for stimulation of prefrontal regions such as the frontal eye fields, we are not aware of evidence suggesting that stimulation of inferior temporal cortex produces non-specific attentional or arousal effects that would bias performance in a complex visual recognition task.
>
> We have now included a short discussion of these potential confounds in Section 4.2.
>
> ### Other points raised by the reviewer
>
> **Limited sample size.**
>
> Non-human primate experiments involving intracranial recordings and causal interventions are inherently challenging. Animals require extensive behavioral training, carefully conducted surgeries to ensure safety and well-being, and are typically shared across multiple ongoing projects, all of which limit their availability for any single study. As a result, it is standard practice in the field to gain statistical power primarily through within-subject factors (e.g., repeated conditions) rather than through large numbers of subjects. This has led to a widely accepted norm of including at least two animals in non-human primate studies for publication in leading journals and conferences. For example, landmark studies such as Bashivan & Kar (Science, 2019, https://doi.org/10.1126/science.aav9436) and Treue & Maunsell (Nature, 1996, https://doi.org/10.1038/382539a0) relied on three and two subjects, respectively, as do many other highly cited primate studies. While ideally we would increase power both within and across subjects, practical and ethical constraints limited us to two animals, consistent with established practice in the field.
>
>
> **In the light of these improvements of the manuscript, we would be grateful to the reviewer if she/he could reassess her/his evaluation of our paper.**

---

### Author Response · Authors · 2025-11-28
**General remark regarding individual responses**

We are very grateful to all 4 reviewers for their critical and constructive evaluation of our work on model-guided microstimulation that steers primate visual behavior. We are addressing almost all points raised by the reviewers below in separate responses. Some of the questions would ideally be answered with additional experimental analysis involving monkey testing. Because the animals’ recording and stimulation devices had to be explanted, the subjects can unfortunately not be used in additional experiments for at least a few months.

To facilitate the comparison between the initial and the current version of the paper, we have underlined all text additions in green.

---

### Author Response · Authors · 2025-12-03
**AC summary - part 1**

We thank all four reviewers for their constructive feedback on our paper “Model-Guided Microstimulation Steers Primate Visual Behavior.” With the help of their comments and suggestions, we were able to substantially improve the paper and to add new analyses directly addressing the points raised. In brief, we thoroughly evaluated model-to-brain alignment, quantified electrode signal quality, added a new diffusion-based visualization with additional controls, and made the manuscript more accessible by better contextualizing the scope. Below we summarize the main changes and clarifications for your consideration. All additions to the initial version of the paper are indicated in the manuscript with a green underline.


**Main paper claims and outputs**

We used topographic vision models to guide the selection of visual stimuli and exact location of stimulation in monkey high-level cortex to guide the animal’s behavior. We observe that our model predicts the behavior of both animals well (**Fig. 3A**) and that our optimization procedure yields a monkey behavioral response with strong effect size (**Fig. 4A**). To provide an intuition of the neural effects of microstimulation, we visualized such activation changes induced by simulated microstimulation using a GAN- and a diffusion-based approach (**Figs. 5 & 11-17**).


## Responses grouped by theme


**Model-brain alignment**

We now describe how model–brain alignment varies along two dimensions: (1) across TDANN variants, and (2) across timepoints. First, we report the full range of cross-validated linear predictivity (explained variance, $R^2$) across models and arrays in **Section 3.1**, showing substantial variance across models and thus motivating the explicit model-selection step – addressing concerns by reviewers **GiCt, LktB, pN2f** regarding the variability / stability, and interpretability of model–brain alignment across models, electrodes, and timepoints. In addition, we trained a non-topographic control model with the same ResNet-18 architecture, objective, dataset, and training schedule but with the spatial loss set to zero, and repeated the same neural-predictivity analysis. Its  $R^2$ values fall within the range spanned by the TDANNs for both monkeys, consistent with the intuition that enforcing smooth topography need not increase (and may slightly trade off) static linear predictivity.

Second, using passive-viewing data recorded four months apart in monkey 1, we show that a mapping trained on the February session predicts June responses as well as (or slightly better than) a mapping trained and tested within June (**Appendix Fig. 10, Section 3.1**; responding to **LktB**). Together, these analyses clarify that (i) model choice matters, (ii) model-brain alignment is stable over months, and (iii) our topographic models are indistinguishable from a non-topographic control with regard to model-brain alignment. The second result of our additional analyses suggest that future experiments may only require passive-viewing data for model–brain mapping every few months rather than a few days before each stimulation session.



**Differences between experiment 1 and 2**

From Exp. 1 to Exp. 2 we relaxed the spatial constraint on candidate stimulation sites to increase the proportion of sites predicted to yield strong effects, rather than simply increasing the number of sites. This change in targeting strategy and the resulting number of tested sites are now described more explicitly in **Section 3.3**. As a result, Exp. 2 produced a significant behavioral bias in the remaining animal (**Fig. 4A**). At the same time, we observed a significant decline in neural signal quality in the 4 months between experiment 1 and 2 using split-half reliability of passive-viewing data, showing a broad, population-level decrease across electrodes (**Appendix Fig. 9, Section 4.2**). These changes directly address the concerns raised by **reviewers GiCt, cYUD, and pN2f** about why experiment 1 showed a strong correlation between model-predicted and observed behavior whereas experiment 2, despite yielding a significant behavioral bias, no longer exhibited a reliable model–behavior relationship.

---

> ### Author Response · Authors · 2025-12-03
> **AC summary - part 2**
>
> **Control conditions**
>
> Reviewers **GiCt, LktB, and pN2f** suggested several valuable baseline conditions—such as random site–sequence pairings, multiple current levels, multi-site stimulation, and perturbation-parameter ablations—which we fully agree would meaningfully strengthen the causal interpretation of model-guided microstimulation. Importantly, both experiments already included matched and interleaved sham and stimulation trials, which served as an internal baseline to prevent expectation effects and address reviewer **GiCt’s** question about whether stimulation was compared against an appropriate non-stimulation control (**Section 3.3**).
> Because neural signal quality deteriorated substantially over the four months between experiments—and after experiment 2 we needed to explant all arrays in both animals (**Appendix Fig. 9**)—we were unable to carry out the additional baselines proposed by the reviewers within the available dataset. In response, we expanded the Discussion (**Section 4.3**) to clarify the rationale for these planned control experiments—1. model-guided vs. random stimulation pairings, 2. multiple current levels, and 3. stimulation using models with deliberately poor neural alignment—and to emphasize them as priority targets for future work.
>
>
> **Visualizations**
>
> We substantially expanded the visualization section by adding a diffusion-based pipeline to the GAN-based approach (**Section 3.4, Appendix A3.1–A3.2**). In response to reviewer **cYUD**, we now describe the architectures, training data, and mapping procedures in detail, and we include a control in which the perturbation vector is randomly permuted before projection into GAN or diffusion latent spaces. For both generative models, targeted TDANN perturbations at highly face-selective sites produce stronger and more systematic face-related changes than permuted controls, especially at moderate current levels (**Figs. 11–17**). At the same time, responding directly to **pN2f**, we explicitly frame these images as intuition-building reconstructions of model IT representations—not as direct readouts of the monkey’s percept.
>
>
> **Scope**
>
> Responding to reviewer **LktB** we clarified that our current claims concern model-guided biasing of recognition behavior in IT. In **Section 4.3 (Discussion)** we now discuss how our approach could generalize to more fine-grained manipulations (e.g., face-feature dimensions within face patches) and how this connects to realistic milestones for prosthetic applications, such as reliably steering category decisions before attempting detailed percept reconstruction.
>
>
> **Overall contribution and impact**
>
> Taken together, the improved manuscript presents an approach to use artificial neural network brain models to guide causal intervention techniques using the example of microstimulation in monkeys performing a visual recognition task. We now provide clearer evidence for four key aspects of the study:
> - **Model selection matters:** different candidate models vary substantially in how well they predict neural data, making an explicit model-selection step essential.
> - **Experiment 1 vs. experiment 2 outcomes:** the differing results across the two experiments are best explained by the pronounced decline in electrode signal quality over the four-month interval, which also limited our ability to run additional control conditions.
> - **Improved visualization analyses:** we expanded our visualization framework by adding a diffusion-based generative model and by including appropriate perturbation controls.
> - **Clarified scope:** we refined the Discussion to make the intended scope, use cases, and limitations of model-guided microstimulation more explicit.
>
> We believe these additions substantially strengthen the paper’s core contribution: a general framework for using topographic vision models to steer microstimulation experiments in a quantitatively testable way. This framework is flexible and can, in principle, be extended to other causal intervention techniques (e.g., focused ultrasound, TMS), to other species including humans, and to other cortical regions.
>
> We further believe that this work will be compelling to a wide audience in both computational neuroscience and machine learning. By showing that vision models can directly inform which neural populations to stimulate—and what behavioral consequences to expect—we provide a concrete demonstration of model-guided causal manipulation in the primate brain. This represents an important step toward integrating artificial neural network models of the brain with causal experimental neuroscience and opens the door to a new class of closed-loop, model-informed perturbation studies.
>
> We kindly ask the Area Chair to consider these substantial improvements and clarifications in the final decision process. For further details on minor revisions and point-by-point replies, we refer to the detailed responses provided below.

---

### Meta-Review · Area_Chair_3sXt · 2025-12-19

**Summary:**

This paper introduces a model-guided microstimulation framework that uses topographic deep artificial neural networks (TDANNs) to predict and steer visual recognition behavior in macaque monkeys. The approach combines brain-aligned topographic models, perturbation modules simulating microstimulation effects, and a mapping procedure to translate model predictions to cortical locations.

Reviewer concerns were:
* Replication and robustness (GiCt, LktB, cYUD): The correlation between model predictions and monkey behavior collapses in Experiment 2, raising questions about the framework's reliability and generalizability.
* Missing control experiments (GiCt, LktB, pN2f): Lack of baselines such as random site-stimulus pairings, multiple current levels, multi-site stimulation, and perturbation parameter ablations.
* Limited sample size (GiCt): Only two monkeys with one excluded from Experiment 2 due to signal degradation.
* Methodological clarity (cYUD, pN2f): Initial manuscript lacked sufficient detail on GAN architecture, regression methods, and electrode-to-model mapping procedures.
* Scope and interpretation (LktB): Need for clearer framing regarding what the approach can realistically achieve (biasing recognition vs. restoring detailed visual percepts).
* Visualization limitations (LktB, cYUD, pN2f): GAN-based visualizations remain qualitative and exploratory, with open questions about their interpretation.

**Reviewer Concerns:**

**Successfully Addressed:**

* Signal quality analysis: Authors now quantify electrode signal degradation between experiments using split-half reliability (Appendix Fig. 9), providing evidence that declining neural signal—rather than model failure—explains the Experiment 2 results.
* Model-brain alignment stability: New analyses demonstrates that model-brain mappings remain stable over 4 months, suggesting alignment procedures need not be repeated frequently.
* Model selection importance: Authors report that linear predictivity (R²) varies substantially across models, justifying the explicit model-selection step.
* Non-topographic control: A control model without spatial loss achieves comparable neural predictivity, clarifying that topography per se doesn't drive brain alignment in this framework.
* Enhanced visualizations: Addition of diffusion-based approach alongside GAN, with random permutation controls demonstrating that targeted perturbations produce stronger category-specific effects than scrambled controls.
* Methodological clarity: Authors expanded descriptions of GAN architectures, training procedures, and mapping methods (Section 3.4, Appendix A3).

**Remaining:**

* Due to array explantation, the authors could not test random site-stimulus pairings, multiple current levels, or deliberately misaligned models.
* Experiment 2 non-replication: While signal degradation provides a reasonable explanation, the lack of model-behavior correlation in Experiment 2 (despite significant behavioral bias) could still raise questions.

**Reviewer Scores:**

* Reviewer GiCt (initial: 4): May have increased to 5-6 given the signal quality analysis and added controls, even though concerns about missing experiments could remain.
* Reviewer LktB (initial: 6): Likely would maintain 6 or possibly increase to 6-7 given the model stability analysis and scope clarifications that directly addressed their concerns.
* Reviewer cYUD (initial: 6): May stay or increase to 7 given the expanded methods section, diffusion-based visualizations, and permutation controls.
* Reviewer pN2f (initial: 8): Likely would maintain 8 as their concerns were well-addressed regarding mapping procedures and signal quality.

---

### Decision · Program_Chairs · 2026-01-26

Accept (Poster)